# Amortized Simulation-Based Inference in Generalized Bayes via Neural Posterior Estimation

**Shiyi Sun**[1]  **Geoff K. Nicholls**[1]  **Jeong Eun Lee**[2]

## Abstract

Generalized Bayesian Inference (GBI) tempers a loss with a temperature $\beta > 0$ to mitigate overconfidence and improve robustness under model misspecification, but existing GBI methods typically rely on costly MCMC or SDE-based samplers and must be re-run for each new dataset and each $\beta$-value. We give the first fully amortized variational approximation for the specific case of the tempered posterior family $p_\beta(\theta\,|\,x) \propto \pi(\theta)p(x\,|\,\theta)^\beta$ by training a single $(x, \beta)$-conditioned neural posterior estimator $q_\phi(\theta \mid x, \beta)$ that enables sampling in a single forward pass, without simulator calls or inference-time MCMC. We introduce two complementary training routes: (i) synthesizes off-manifold samples $(\theta, x) \sim \pi(\theta)p(x \mid \theta)^\beta$ and (ii) reweights a fixed base dataset $\pi(\theta)p(x \mid \theta)$ using self-normalized importance sampling (SNIS), where we show that the SNIS-weighted objective provides a consistent forward-KL fit to the tempered posterior with finite weight variance. Across four standard simulation-based inference (SBI) benchmarks—including the chaotic Lorenz–96 system—our $\beta$-amortized estimator achieves competitive posterior approximations, in standard two-sample metrics, with non-amortized MCMC-based power-posterior samplers over a wide range of temperatures. Code is available at https://github.com/Komorebiww/amortized-generalized-bayes.

## 1. Introduction

Simulation-based inference (SBI) (Cranmer et al., 2020) addresses Bayesian inference when the likelihood is implicit,

[1]Department of Statistics, University of Oxford, Oxford, United Kingdom [2]Department of Statistics, University of Auckland, Auckland, New Zealand. Correspondence to: Shiyi Sun <shiyi.sun@spc.ox.ac.uk>.

*Proceedings of the 43rd International Conference on Machine Learning*, Seoul, South Korea. PMLR 306, 2026. Copyright 2026 by the author(s).

yet a simulator can generate samples. Given parameters $\theta$, one can simulate $x \sim p(x \mid \theta)$ even though the likelihood $p(x \mid \theta)$ is intractable (Lueckmann et al., 2021). This setting is common in scientific and engineering domains where complex mechanistic models can produce synthetic data, but evaluating or differentiating the likelihood is infeasible (Alsing et al., 2019; Lueckmann et al., 2017). Methods span from approximate Bayesian computation (ABC) (Beaumont et al., 2002; Marjoram et al., 2003) to neural posterior estimation (NPE) (Greenberg et al., 2019; Papamakarios & Murray, 2016) and neural likelihood/ratio estimation (NLE/NRE) (Papamakarios et al., 2019; Hermans et al., 2020; Durkan et al., 2020; Thomas et al., 2022). Many modern approaches further employ amortized inference (Greenberg et al., 2019; Papamakarios et al., 2019; Durkan et al., 2020; Gonçalves et al., 2020; Radev et al., 2020; Elsemüller et al., 2024), training neural networks that map simulated data directly to posterior or likelihood surrogates. Amortization enables efficient reuse across multiple observations and substantially lowers per-dataset inference cost compared with classical sampling-based methods (Lueckmann et al., 2021), but it typically presumes a well-specified simulator and that targeting the exact Bayesian posterior is desirable.

However, in practice, simulators are imperfect representations of reality: model misspecification and measurement noise can cause the Bayesian posterior to concentrate on pseudo-true regions that need not correspond to meaningful or stable solutions (Berk, 1966; Kleijn & van der Vaart, 2006; Grünwald & Van Ommen, 2017). To address such issues, generalized Bayesian inference (GBI) (Bissiri et al., 2016) updates beliefs using a loss-based surrogate in place of the likelihood. This approach can be rigorously justified through the lens of optimization-centric variational inference (Knoblauch et al., 2022), providing a principled route to robustness. Concretely, following (Bissiri et al., 2016), we write

$$L(\theta; x_{\mathrm{obs}}) \equiv \exp\{-\beta\,l(\theta; x_{\mathrm{obs}})\},$$

where $l(\theta; x_{\mathrm{obs}})$ measures the quality of parameter $\theta$ relative to the observation and $\beta > 0$ is a temperature that balances data fit against the prior. With a prior $\pi(\theta)$ and simulator-defined likelihood $p(x \mid \theta)$, the corresponding generalized

posterior is

$$p(\theta \mid x_{\text{obs}}) \propto \exp\{-\beta\, l(\theta; x_{\text{obs}})\}\, \pi(\theta). \qquad (1)$$

Choosing $l(\theta; x_{\text{obs}}) = -\log p(x_{\text{obs}} \mid \theta)$ yields the tempered (power) posterior (Friel & Pettitt, 2008; Marinari & Parisi, 1992), which preserves the likelihood-based structure while introducing a tunable robustness via $\beta$ (down-weighting data for $\beta < 1$, up-weighting for $\beta > 1$). Related robustness perspectives include (Holmes & Walker, 2017; Lyddon et al., 2019), and practical choices of $\beta$ can be calibrated to achieve nominal frequentist properties (Syring & Martin, 2019).

While generalized bayes provides a principled route to robustness, applying it to simulator-based settings remains challenging: evaluating the loss or scoring-rule expectations typically requires many simulator queries at inference time. Two recent strands address this gap. (i) Scoring-rule posteriors: Pacchiardi et al. (2024) replace the likelihood by strictly proper scoring rules (e.g., energy and kernel scores) to obtain a generalized posterior with consistency and outlier-robustness guarantees; inference is performed via pseudo-marginal MCMC or stochastic-gradient MCMC and is not amortized across observations. (ii) Amortized cost estimation (ACE): Gao et al. (2023) train a neural network to approximate the expected cost used in the GBI update, thereby amortizing cost evaluation and avoiding simulator calls at test time; samples from the generalized posterior are then obtained with MCMC over parameters. Both strands reduce reliance on explicit likelihoods but still require inference-time sampling over $\theta$, which limits throughput when many $x_{\text{obs}}$ must be processed or computational resources are constrained. This motivates our approach of directly amortizing the family of power posteriors $p_\beta(\theta \mid x)$ with NPE.

**Contribution.** Building on generalized Bayes, we focus on the power posterior

$$p_\beta(\theta \mid x) \propto \pi(\theta)\, p(x \mid \theta)^\beta, \qquad (2)$$

which interpolates between the prior ($\beta \to 0$) and the standard posterior ($\beta = 1$). Whereas existing approaches typically require inference-time MCMC or SDE/score-based samplers to draw from $p_\beta(\theta \mid x)$, we propose a fully amortized alternative over both $x$ and $\beta$ that directly learns the family of power posteriors via neural posterior estimation (NPE). Concretely, we train a $\beta$-conditioned estimator $q_\phi(\theta \mid x, \beta)$ so that sampling from $p_\beta(\theta \mid x)$ at test time reduces to a single forward pass, without simulator calls or any MCMC. After training, users can sweep $\beta$ to inspect posterior stability, posterior-predictive behavior, and ESS diagnostics, or plug the amortized sampler into existing $\beta$-calibration procedures without rerunning inference for each temperature.

We instantiate this idea through two complementary training routes:

- **Route A — Tempered-joint synthesis.** We build an offline dataset of $(\theta, x, \beta)$ by approximately sampling from the (unnormalized) tempered joint density, $p_\beta(\theta, x) \propto \pi(\theta)\, p(x \mid \theta)^\beta$. To approximate this joint, we learn the joint score $\nabla_{(\theta, x)} \log\{\pi(\theta)p(x \mid \theta)^\beta\}$ and use short-run Langevin dynamics across a temperature schedule to synthesize pairs. Training NPE on these pairs lets the amortized model directly learn $p_\beta(\theta \mid x)$.

- **Route B — SNIS-weighted NPE (global per-$\beta$ normalization).** Draw a base dataset $(\theta_i, x_i)_{i=1}^N \sim \pi(\theta)p(x \mid \theta)$ once and reuse it across temperatures. For any $m(x) > 0$, define the tempered joint $\pi_\beta(\theta, x) \propto \pi(\theta)\, p(x \mid \theta)^\beta m(x)$, whose conditional equals $p_\beta(\theta \mid x)$. Using the base joint as proposal, the self-normalized importance weight is

$$w_\beta(\theta, x) = \frac{\pi_\beta(\theta, x)}{\pi(\theta)p(x \mid \theta)} = p(x \mid \theta)^{\beta-1}m(x).$$

We fit a single amortized $q_\phi(\theta \mid x, \beta)$ by a per-temperature SNIS-weighted NPE objective.

We then minimize the objective

$$\mathcal{L}_{\text{SNIS}}(\phi) = -\sum_{\beta \in \mathcal{B}} \sum_{i=1}^N \tilde{w}_{\beta,i} \log q_\phi(\theta_i \mid x_i, \beta).$$

where $\tilde{w}_{\beta,i} \propto w_\beta(\theta_i, x_i)$. Two practical instantiations are: (i) NLE-based with $m(x) = 1$, plugging a learned $\hat{p}_\eta(x \mid \theta)$ for $p(x \mid \theta)$; (ii) NRE-based with $m(x) = p(x)^{1-\beta}$, where a classifier ratio $\hat{r}_\psi(x \mid \theta) \approx p(x \mid \theta)/p(x)$ yields $w_\beta(\theta, x) \propto \hat{r}_\psi(\theta, x)^{\beta-1}$.

Route B resembles Sequential Neural Variational Inference (SNVI) (Glöckler et al., 2022): SNVI makes a variational NPE-approximation to a posterior; the likelihood used to form the target-posterior is an NLE-approximation made using SBI. The NPE is not amortized. We target GBI and amortize data and $\beta$ but similarly have an NPE objective targeting an NLE approximation.

## 2. Problem Setup

Following the fully amortized objective introduced above, we now formalize the learning problem. Let $\Theta \subseteq \mathbb{R}^{d_\theta}$ and $\mathcal{X} \subseteq \mathbb{R}^{d_x}$ be measurable spaces, $\pi(\theta)$ a prior, and $x \sim p(\cdot \mid \theta)$ an implicit simulator. For $\beta \in \mathcal{B}$, define the power posterior in (2) with normalizer $Z_\beta(x) = \int_\Theta \pi(\theta)\, p(x \mid \theta)^\beta\, \mathrm{d}\theta < \infty$. Our goal is to learn a single amortized estimator $q_\phi(\theta \mid x, \beta)$ that approximates $p_\beta(\theta \mid x)$ for any $\beta$ in a prescribed set $\mathcal{B}$ (a discrete grid or a closed interval). At test time, $\beta$ is user-specified; during training, $\beta \sim p(\beta)$ supported on $\mathcal{B}$ to ensure coverage.

**Temperature constraints.** As $\beta \downarrow 0$, $p_\beta(\theta \mid x) \to \pi(\theta)$; at $\beta = 1$ we recover standard Bayes; for $0 < \beta < 1$ the likelihood is down-weighted (typically improving robustness under misspecification), while $\beta > 1$ up-weights the likelihood and often yields tighter posteriors under well-specified simulators. To avoid degeneracy and guarantee $Z_\beta(x) < \infty$, we restrict $\beta$ to a bounded domain $\mathcal{B} = [\beta_{\min}, \beta_{\max}]$ (or its grid) with $0 < \beta_{\min} \le 1 \le \beta_{\max} < \infty$, chosen for numerical stability.

## 3. Related work

**Neural estimators for SBI.** Modern SBI amortizes either the posterior, likelihood, or likelihood ratio from simulated pairs (Papamakarios & Murray, 2016; Greenberg et al., 2019; Papamakarios et al., 2019; Hermans et al., 2020; Durkan et al., 2020; Thomas et al., 2022). These approaches enable fast reuse across observations but typically target the standard Bayesian posterior ($\beta = 1$); recent studies examine calibration/robustness under misspecification (Cannon et al., 2022).

**Generalized Bayes in SBI.** Generalized Bayes replaces the likelihood by a loss-based update with a temperature $\beta$ (Bissiri et al., 2016; Holmes & Walker, 2017; Lyddon et al., 2019). Within SBI, Gao et al. (2023) amortize the expected cost (ACE) and subsequently draw parameter samples via MCMC for each data set $x_0$ at test time. Algorithm 1 provides a brief summary of their procedure. Pacchiardi et al. (2024) instead construct scoring-rule posteriors using strictly proper scoring rules (e.g., energy or kernel scores) and perform pseudo-marginal or SG-MCMC inference, but their approach is not amortized at test time. Carmona & Nicholls (2022) and Battaglia et al. (2025) amortize GBI *hyperparameters* using conditional normalizing flows and reverse-KL (rKL) variational inference, pointing to the difficulty of amortizing over data in GBI as a reason for relying on rKL minimization. In contrast, our work directly amortizes the family $p_\beta(\theta \mid x)$ across both observations $x$ and the GBI hyperparameter $\beta$ within the SBI framework, thereby avoiding both test-time MCMC and retraining.

---

**Algorithm 1** Gao et al. (2023): Generalized Bayes via ACE + MCMC

**Inputs:** prior $\pi(\theta)$, simulator $p(x \mid \theta)$, loss $\ell$, temperature $\beta$.

1: Learn $C_\phi(\theta, x) \approx \mathbb{E}_{x' \sim p(\cdot|\theta)}[\ell(x', x)]$ ($x$-amortized surrogate) by regression on pairs $(\theta, x)$ (ACE).
2: Define $\log \tilde{p}_\beta(\theta \mid x_0) = \log \pi(\theta) - \beta C_\phi(\theta, x_0)$ (unnormalized).
3: Run MCMC targeting $\tilde{p}_\beta(\theta \mid x_0)$ to draw $\{\theta^{(s)}\}$.

---

**Score-based synthesis.** Score-based models learn the score $\nabla \log p$ via denoising score matching (Hyvärinen, 2005; Vincent, 2011; Song & Ermon, 2019). A closely related alternative is to learn a conditional posterior score $s_\psi(\theta, t; x) \approx \nabla_\theta \log p_t(\theta \mid x)$ (equivalently, a conditional diffusion model in $\theta$ given $(x, \beta)$), so that score evaluation amortizes jointly over observations and inverse temperatures. However, unlike a one-shot posterior sampler, generating samples at test time still requires running an iterative reverse-time procedure—e.g., predictor–corrector sampling or short-run Langevin dynamics—for each new observation $x$ (and each $\beta$) (Geffner et al., 2023; Sharrock et al., 2022), which can dominate inference latency.

Route A (Section 4.1) uses a learned joint score over $(\theta, x)$ purely as an offline generator (short-run Langevin across $\beta$) to create tempered training pairs, while inference uses a standard amortized posterior network. While we implement NCSN for concreteness, the route is agnostic to the scoring paradigm: VE/VP score-SDE frameworks (Song et al., 2020) and EDM (Karras et al., 2022) can be dropped in as alternatives (replacing the noise scale by a time parameter), with predictor–corrector or short-run Langevin samplers yielding the same NPE training objective.

**IS-aware/forward-KL objectives.** For a fixed temperature $\beta$, our objective

$$\mathcal{L}_\beta(\phi) = \mathbb{E}_{(\theta, x) \sim \pi_\beta}[-\log q_\phi(\theta \mid x, \beta)]$$

is a forward-KL (fKL) fit $\mathbb{E}_x \mathrm{KL}(p_\beta(\theta \mid x) \| q_\phi(\theta \mid x, \beta))$—a mass-covering criterion (Minka, 2005; Hernandez-Lobato et al., 2016; Li & Turner, 2016). Because $\pi_\beta(\theta, x) \propto \pi(\theta) \, p(x \mid \theta)^\beta \, m(x)$ can't be sampled directly, we estimate $\mathcal{L}_\beta$ with self-normalized importance sampling (SNIS) (Murphy, 2012), using the joint $\pi(\theta)p(x \mid \theta)$ as proposal. The resulting weighted MLE, $\widehat{\mathcal{L}}_\beta(\phi) = \sum_i \tilde{w}_{i,\beta}[-\log q_\phi(\theta_i \mid x_i, \beta)]$, has gradients that equal the SNIS estimate of the fKL gradient. This connects Route B to a long line of work on proposal learning by fKL and SNIS, including adaptive/mixture-boosting IS (Cappé et al., 2008; Cornuet et al., 2012; Jerfel et al., 2021) and the wake/sleep family where the inference network $q$ is trained by $\mathrm{KL}(p\|q)$ using (re)weighted samples (Hinton et al., 1995; Bornschein & Bengio, 2014). In our setting, the "proposal" being refined is the conditional $q_\phi(\theta \mid x, \beta)$ (e.g., a Mixture Density Network), giving an amortized fKL refinement across both $x$ and $\beta$.

## 4. Amortized Power Posteriors via NPE

We amortize the family of power posteriors (2) by training a single $\beta$-conditioned NPE $q_\phi(\theta \mid x, \beta)$, enabling single-pass sampling for any $(x, \beta)$ without simulator calls or inference-time MCMC. We realize this via two routes: (A) score-assisted synthesis of tempered triplets $(\theta, x, \beta)$; (B) reuse of

a base joint dataset with per-$\beta$ self-normalized importance weights (NLE/NRE instantiations). Implementation details and trade-offs follow.

## 4.1. Route A: score-assisted tempered synthesis + NPE

---

**Algorithm 2** Route A: score-assisted tempered synthesis + NPE

---

**Require:** prior $\pi(\theta)$, simulator $p(x \mid \theta)$, annealed noise scales $\sigma_1 > \cdots > \sigma_T$, base step-size scale $\epsilon_\eta$, $\beta$-grid $\mathcal{G}$ defining $p(\beta)$

**Ensure:** amortized posterior $q_\phi(\theta \mid x, \beta)$
1: **Phase I: learn joint score $s_\psi$ (DSM).**
2: **for** minibatches $(\theta, x) \sim \pi(\theta)p(x \mid \theta)$ **do**
3:     sample $\sigma \in \{\sigma_t\}, \epsilon \sim \mathcal{N}(0, I)$; set $(\tilde{\theta}, \tilde{x}) = (\theta, x) + \sigma\epsilon$
4:     update $\psi$ by minimizing $\| s_\psi(\tilde{\theta}, \tilde{x}, \sigma) + \epsilon/\sigma \|^2 \times \frac{1}{\sigma}$
5: **end for**
6: **Phase II: synthesize tempered pairs (short-run Langevin).**
7: **for** $\beta \sim p(\beta)$ **do**
8:     initialize $(\theta, x) \sim \pi(\theta)p(x \mid \theta)$
9:     **for** $j = 1, \dots, T$ **(anneal over noise scales) do**
10:       $\eta_j \leftarrow \epsilon_\eta \sigma_j^2/\sigma_T^2$    $\eta_j$ is fixed for this noise scale
11:       **for** $k = 0, \dots, K_j - 1$ **do**
12:         $g_{j,k} \leftarrow \beta s_\psi(\theta, x, \sigma_j) - (\beta - 1)(s_\pi(\theta), 0)$
          with $s_\pi = \nabla_\theta \log \pi$
13:         $(\theta, x) \leftarrow (\theta, x) + \eta_j g_{j,k} + \sqrt{2\eta_j} \xi_{j,k}$,
          $\xi_{j,k} \sim \mathcal{N}(0, I)$
14:       **end for**
15:     **end for**
16:     add $(\theta, x, \beta)$ to dataset $\mathcal{D}$
17: **end for**
18: **Phase III: Train NPE.**
19: Minimize $\min_\phi \mathbb{E}_{(\theta, x, \beta) \in \mathcal{D}}\big[ -\log q_\phi(\theta \mid x, \beta)\big]$

---

A natural way to train a NPE targeting power posterior (2) is to construct a training set of $(\theta, x, \beta)$ triples which is distributed as the tempered joint

$$\tilde{p}_\beta(\theta, x) \propto \pi(\theta)p(x|\theta)^\beta \qquad (3)$$

when conditioned on $\beta$. If $c_\beta(\theta) = \int p(x|\theta)^\beta dx$ then the marginal, $\tilde{p}_\beta(\theta) \propto \pi(\theta)c_\beta(\theta)$, is not the prior (except at $\beta = 1$) and $\tilde{p}_\beta(x|\theta) = p(x|\theta)^\beta/c_\beta(\theta)$ has unwanted $\theta$ and $\beta$ dependence, but $c_\beta(\theta)$ cancels in the joint $\tilde{p}_\beta(\theta)\tilde{p}_\beta(x|\theta)$, so MCMC targeting (3) gives pairs $(\theta, x)$ with the property that $\theta|x \sim p_\beta(\theta|x)$ in (2).

We first learn a joint score $s_\psi(\theta, x) \approx \nabla_{(\theta,x)} \log p(\theta, x)$ using NCSN (Song & Ermon, 2019) on samples from the base joint distribution $p(\theta, x) = \pi(\theta)p(x \mid \theta)$. We then synthesize tempered pairs by running Langevin dynamics targeting tempered joint distribution $\tilde{p}_\beta(\theta, x)$:

$$(\theta^{k+1}, x^{k+1}) \leftarrow (\theta^k, x^k) + \eta \underbrace{\nabla_{(\theta,x)} \log \tilde{p}_\beta(\theta^k, x^k)}_{\text{use score + prior + } \beta}$$
$$+ \sqrt{2\eta}\, \xi^k. \qquad (4)$$

with $\xi^k \sim \mathcal{N}(0, I)$ and a step size $\eta$ chosen according

to the annealed noise scale. The joint score $s(\theta, x) = \nabla_{(\theta,x)} \log p(\theta, x)$ is

$$\nabla_{(\theta,x)} \log p(\theta, x) = \nabla_{(\theta,x)} \log \pi(\theta) + \nabla_{(\theta,x)} \log p(x \mid \theta)$$
$$= \begin{pmatrix} \nabla_\theta \log \pi(\theta) \\ \nabla_x \log \pi(\theta) \end{pmatrix} + \begin{pmatrix} \nabla_\theta \log p(x \mid \theta) \\ \nabla_x \log p(x \mid \theta) \end{pmatrix}. \qquad (5)$$

To obtain $\nabla_{(\theta,x)} \log \tilde{p}_\beta(\theta, x)$, we can rewrite it in the following way:

$$\nabla_{(\theta,x)} \log \tilde{p}_\beta(\theta, x) = \nabla_{(\theta,x)} \log \pi(\theta) + \beta \nabla_{(\theta,x)} \log p(x \mid \theta)$$
$$= \beta \nabla_{(\theta,x)} \log p(\theta, x)$$
$$- (\beta - 1)(\nabla_\theta \log \pi(\theta), \mathbf{0})$$
$$\approx \beta s_\psi(\theta, x) - (\beta - 1)(\nabla_\theta \log \pi(\theta), \mathbf{0}). \qquad (6)$$

where in the first equality we substitute (3), in the second we note $\nabla_x \log \pi(\theta) = 0$ and apply Bayes' rule and finally $s_\psi(\theta, x) \approx s(\theta, x)$ from Algorithm 2.

**Training objective.** Once a large training set $\mathcal{D}_\beta = \{(\theta, x)\}$ is generated for multiple $\beta$ values (sampled from some $p(\beta)$), we fit $q_\phi$ as a conditional MLE:

$$\min_\phi \mathbb{E}_{\beta \sim p(\beta)}\mathbb{E}_{(\theta,x) \sim \mathcal{D}_\beta}\Big[ -\log q_\phi(\theta \mid x, \beta)\Big]. \qquad (7)$$

The algorithm for Route A is given in Algorithm 2, more details about the selection of hyperparameters are in Appendix A.1.

## 4.2. Route B: Using SNIS to reweight the NPE (no scores, no MCMC)

On the joint space, consider the tempered family

$$\pi_\beta(\theta, x) \propto \pi(\theta) p(x \mid \theta)^\beta m(x), \qquad m(x) > 0, \quad (8)$$

for which the conditional is unchanged for any $m(x)$: $\pi_\beta(\theta \mid x) = p_\beta(\theta \mid x) \propto \pi(\theta) p(x \mid \theta)^\beta$. Here $m(x)$ will be used to accommodate different likelihood estimators. To amortize $p_\beta(\theta \mid x)$ with a $\beta$-conditioned NPE $q_\phi(\theta \mid x, \beta)$, we would like to minimize

$$\mathcal{L}_\phi := \mathbb{E}_{(\theta,x) \sim \pi_\beta}\big[ -\log q_\phi(\theta \mid x, \beta)\big]. \qquad (9)$$

Sampling from $\pi_\beta(\theta, x)$ is not possible, so we use self-normalized importance sampling (SNIS) (Murphy, 2012) to reweight the base joint $p(\theta, x) = \pi(\theta)p(x \mid \theta)$. The unnormalized joint importance weight is

$$w_\beta(\theta, x) = \frac{\pi(\theta) p(x \mid \theta)^\beta m(x)}{\pi(\theta) p(x \mid \theta)} = p(x \mid \theta)^{\beta-1} m(x). \qquad (10)$$

**Two practical choices for $m(x)$.**

- $m(x) = 1$ (**NLE-based**): $w_\beta(\theta, x) = p(x \mid \theta)^{\beta-1}$, approximated by $\hat{p}_\eta(x \mid \theta)$, a conditional likelihood trained using an NLE (Papamakarios et al., 2019).

- $m(x) = p(x)^{1-\beta}$ (**NRE-based**): the weights are $w_\beta(\theta, x) = \left(\frac{p(x|\theta)}{p(x)}\right)^{\beta-1}$, which can be acquired from a classifier-based likelihood-ratio estimator $\hat{r}_\psi(x \mid \theta)$ (NRE) via $\hat{r}_\psi = \frac{d_\psi}{1-d_\psi}$ with equal class priors (Hermans et al., 2020).

In both cases, we normalize the weights $\tilde{w}_\beta(\theta, x) = \frac{w_\beta(\theta,x)}{\sum w_\beta(\theta,x)}$. By the self-normalized importance sampling identity (Murphy, 2012), the objective is

$$\mathcal{L}_\phi = \frac{\mathbb{E}_{(\theta,x)\sim p(x,\theta)}\left[w_\beta(\theta, x)\left(-\log q_\phi(\theta \mid x, \beta)\right)\right]}{\mathbb{E}_{(\theta,x)\sim p(x,\theta)}\left[w_\beta(\theta, x)\right]}. \quad (11)$$

We estimate (11) with

$$\begin{aligned}\widehat{\mathcal{L}}_\phi(\beta) &= \frac{\sum_{i=1}^N w_{\beta,i}\left(-\log q_\phi(\theta_i \mid x_i, \beta)\right)}{\sum_{i=1}^N w_{\beta,i}} \\ &= \sum_{i=1}^N \tilde{w}_{\beta,i}\left(-\log q_\phi(\theta_i \mid x_i, \beta)\right),\end{aligned} \quad (12)$$

where $\tilde{w}_{\beta,i} := \frac{w_{\beta,i}}{\sum_{j=1}^N w_{\beta,j}}$.

Minimizing (11) is equivalent to minimizing the fKL from the tempered posterior to $q_\phi$, formalized below.

**Proposition 4.1.** *For any $\beta \in \mathcal{B}$,*

$$\arg\min_\phi \mathcal{L}_\phi = \arg\min_\phi \mathbb{E}_x\left[\text{KL}\left(p_\beta(\theta \mid x) \,\|\, q_\phi(\theta \mid x, \beta)\right)\right].$$

The full proof is given in Appendix B.

In importance sampling, the dispersion of the weights is critical. For the NRE-based construction, with weights defined in (10), we establish the following.

**Proposition 4.2.** *The variance of the NRE weights is finite for $\beta \in [1/2, 1]$:*

$$\text{Var}\left[w_\beta\right] < 1.$$

The statement and proof are given in Appendix B. When we estimate the loss in (12) the weights are independent over $i = 1, \ldots, n$. However we recycle the samples $\theta_i$ and $x_i$ over different $\beta$-values. This doesn't break the independence assumption needed for Proposition 4.2 as that result conditions on a single fixed $\beta$.

Both choices (NLE-based and NRE-based) instantiate the same SNVI target $\pi_\beta(\theta \mid x)$. We implement both and compare their performance on different benchmarks. Algorithm 3 gives the NRE-based SNIS procedure; the NLE-based variant is analogous, replacing the ratio estimator with a neural likelihood estimator. The neural architectures are given in Appendix A.2.

---

**Algorithm 3** Route B: SNIS-weighted NPE (NRE-based)

---

**Require:** prior $\pi(\theta)$, simulator $p(x \mid \theta)$, $\beta$-grid $\mathcal{G}$, ratio net $d_\psi$, NPE $q_\phi$
**Ensure:** amortized posterior $q_\phi(\theta \mid x, \beta)$
1: **Base data:** draw $(\theta_i, x_i) \sim \pi(\theta)p(x \mid \theta)$ and negatives $(\tilde{\theta}_i, \tilde{x}_i) \sim \pi(\theta)p(x)$ for $i = 1, \ldots, n$
2: **Train ratio (NRE-B):** minimize $\mathcal{L}_\psi = \frac{1}{n}\sum_i\left[-\log d_\psi(\theta_i, x_i) - \log(1 - d_\psi(\tilde{\theta}_i, \tilde{x}_i))\right]$
3: compute $s_i = \log \hat{r}_\psi(x_i \mid \theta_i)$ with $\hat{r}_\psi = \frac{d_\psi}{1 - d_\psi}$
4: **for** $\beta \in \mathcal{G}$ **do** {global log-normalizer per temperature}
5: $\quad S_\beta \leftarrow \log \sum_{i=1}^n \exp\left((\beta - 1)s_i\right)$
6: $\quad\quad$ set $\tilde{w}_{\beta,i} = \exp\left((\beta - 1)s_i - S_\beta\right)$ for $i = 1, \ldots, n$
7: **end for**
8: **Train NPE (SNIS weights):**
9: $\quad$ for $\beta \in \mathcal{G}$ and mini-batches $\mathcal{I} \subset \{1, \ldots, n\}$, minimize $\mathcal{L}_\phi = \frac{n}{|\mathcal{I}|}\sum_{i \in \mathcal{I}}\tilde{w}_{\beta,i}\left[-\log q_\phi(\theta_i \mid x_i, \beta)\right]$[1]

---

## 5. Experiments

### 5.1. Benchmark

We evaluate $\beta$-amortized posterior inference by comparing samples from a single $\beta$-conditioned estimator $q_\phi(\theta \mid x, \beta)$ to reference samples from the corresponding power posterior $p_\beta(\theta \mid x) \propto \pi(\theta)p(x \mid \theta)^\beta$.

Experiments are conducted on four standard SBI benchmarks (Lueckmann et al., 2021; Lorenz, 1996) (Gaussian Mixture, Two Moons, SLCP, and Lorenz–96), chosen to span increasing posterior complexity from low-dimensional multimodality to chaotic dynamics. We report sample-based two-sample discrepancies (MMD (Gretton et al., 2012) and C2ST (Friedman, 2004; Lopez-Paz & Oquab, 2016)) as functions of $\beta$, and study the effect of simulation budget (10k vs. 100k) to quantify both temperature-robustness and data-efficiency. We further compare Route A (score-assisted tempered-pair synthesis) and Route B (SNIS-weighted NPE reweighting) to evaluate approximation fidelity across amortization budgets and design choices.

**Evaluation protocol.** For each benchmark, a posterior evaluation is conditioned on a single held-out observation $x_{\text{obs}}$; a task refers to one such observation together with its reference power posterior. The curves in Figure 1 are averaged over held-out tasks rather than being produced from a single favorable observation. The symbol $n$ in the figure denotes the simulation budget, i.e., the number of simulator-generated training pairs used to fit the amortized estimator, not the number of datapoints inside an observation. Although $q_\phi(\theta \mid x, \beta)$ treats $\beta$ as an input, we report results on a finite evaluation grid of $\beta$ values so that amortized

---

[1]The mini-batch gradient in Step 9 is an unbiased estimator of the full-data weighted objective. A complete proof is given in Appendix B.

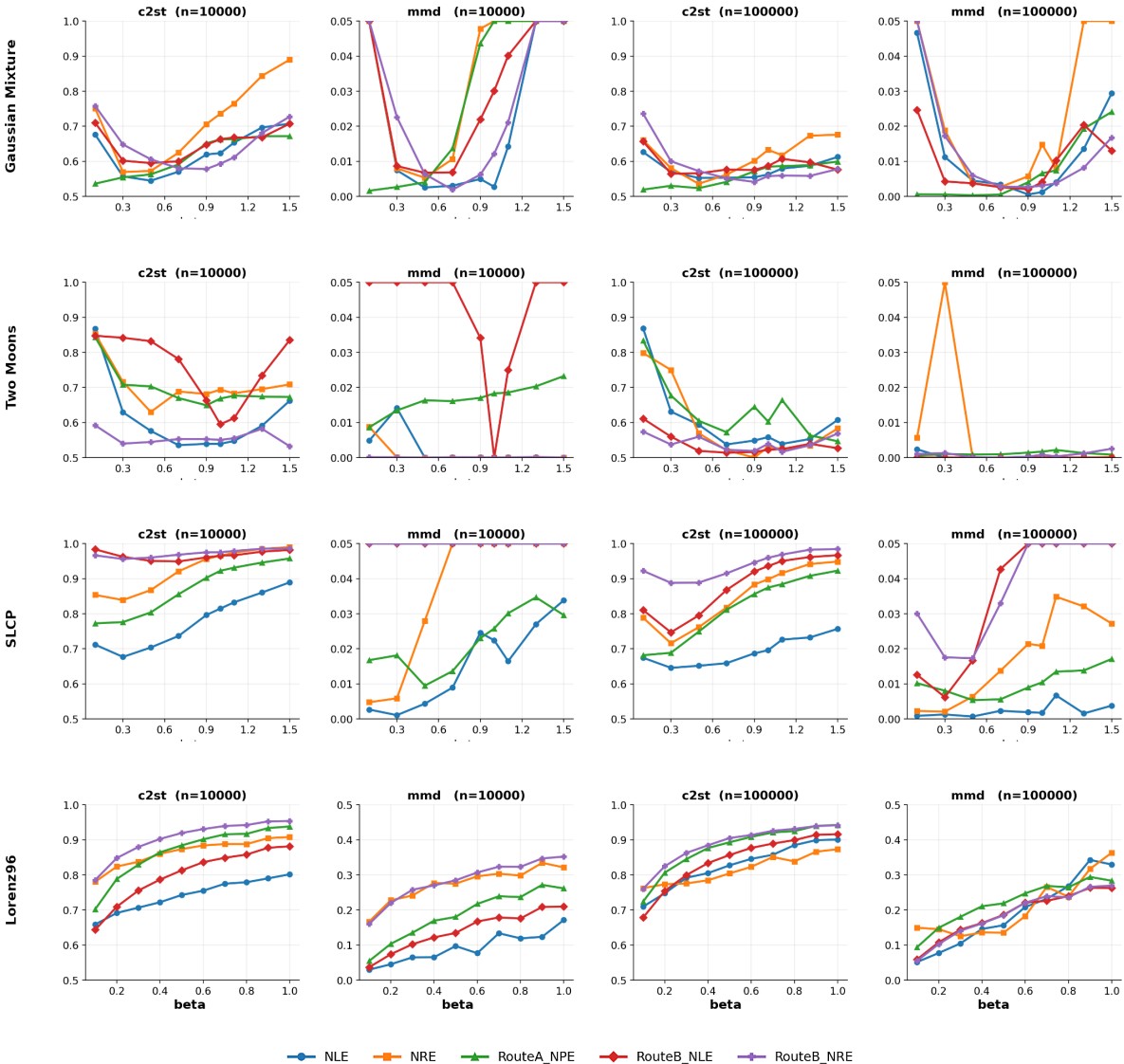

*Figure 1.* Route A/B comparison across benchmarks with a shared legend.

samples can be compared directly with reference samplers at the same temperatures. Full benchmark definitions, reference samplers, hyperparameters, and evaluation details are provided in Appendix C and Appendix E.

We do not claim that the amortized methods are uniformly best. However, Figure 1 shows that our $\beta$-conditioned estimator $q_\phi(\theta \mid x, \beta)$ achieves competitive approximation quality to the reference power posteriors $p_\beta(\theta \mid x)$ without inference-time MCMC. The results should be interpreted through the two complementary diagnostics: MMD captures distributional discrepancies in kernel mean embeddings, while C2ST is often more sensitive to small but classifiable differences between sample sets. Performance tends to be better in regimes where the tempered target remains well

covered by the training distribution, and it can degrade in harder benchmarks or at extreme $\beta$ values where either the importance weights become concentrated or score-based synthesis becomes less stable. **Route B** (SNIS-weighted NPE) can be an efficient choice when $\beta$ is not far from $1$ (especially for $\beta \leq 1$), where reweighting is comparatively stable. **Route A** (score-assisted synthesis) can be advantageous when SNIS reweighting becomes unreliable (e.g., ratio/likelihood miscalibration or ESS collapse at small $\beta$) or when the base joint undercovers tempered regions; it may improve support coverage at the cost of additional offline compute and potential bias from score error and short-run Langevin dynamics. Appendix A.1 includes a step-size ablation on Gaussian mixture, illustrating the sensitivity of Route A synthesis to the Langevin discretization. For addi-

tional intuition on how $\beta$ reshapes the posterior geometry, see Appendix G (Figs. 6–7).

Route A appears relatively robust at small $\beta$ in the Gaussian mixture and SLCP tests, where the weight variance for Route B SNIS methods increases. At $\beta < 1$, SNIS-weighted NPE approximates a diffuse target using samples from a more concentrated base joint, so the Route B curves can rise in both C2ST and MMD as $\beta$ decreases. This interpretation is supported by the Route B effective-sample-size diagnostics in Appendix D, where ESS is highest near $\beta = 1$ and drops as $\beta$ moves away from the base proposal, especially for SLCP and Lorenz–96. The methods generally show higher discrepancies on more structured posterior geometries and in challenging temperature regimes. Route A generates training samples via short-run annealed Langevin dynamics, which can help with support exploration but is sensitive to score error and multimodal geometry, as seen in Two Moons. Overall, the amortized methods are broadly comparable to the non-amortized references while making posterior queries for many $\beta$ values substantially cheaper after training.

### 5.2. Single-Compartment Hodgkin–Huxley

We evaluate our method on a challenging scientific simulator, the single-compartment Hodgkin–Huxley (HH) model of neuronal voltage dynamics (Teeter et al., 2018; Pospischil et al., 2008). This example is not a ground-truth recovery benchmark but a first step toward a more realistic, computationally expensive, misspecified scientific-simulator setting, where amortization is practically useful. Following prior work (Gonçalves et al., 2020), the simulator has eight parameters and we use seven summary statistics computed from the voltage trace as observations. Full HH model details (parameter definitions, priors, and summary statistics) are provided in Appendix F. We perform inference for real electrophysiological recordings from the Allen Cell Types Database (Allen Institute for Brain Science, 2016), a setting known to be difficult due to model mismatch and experimental variability(Gao et al., 2023; Tolley et al., 2024).

We train our $\beta$-conditional posterior estimator using `RouteB_NLE` with 10,000 prior-sampled simulations. Figure 2 shows the one-dimensional marginal posteriors for the eight HH parameters under $\beta \in \{0.1, 1.0, 2.0\}$. The marginals are broadly consistent across $\beta$, with only modest shifts in a subset of parameters (visible changes in the tails/peaks for $g_{\text{Na}}$ and $g_{\text{K}}$) while others remain nearly unchanged ($E_{\text{leak}}$). Figure 3 provides a posterior predictive check on three selected experimental sweeps: simulations generated from samples of $q_\phi(\theta \mid x_{\text{obs}}, \beta)$ at $\beta = 0.1$ qualitatively reproduce the observed spiking patterns, with predicted traces capturing the main spike timing structure across the three observations. Overall, these results indi-

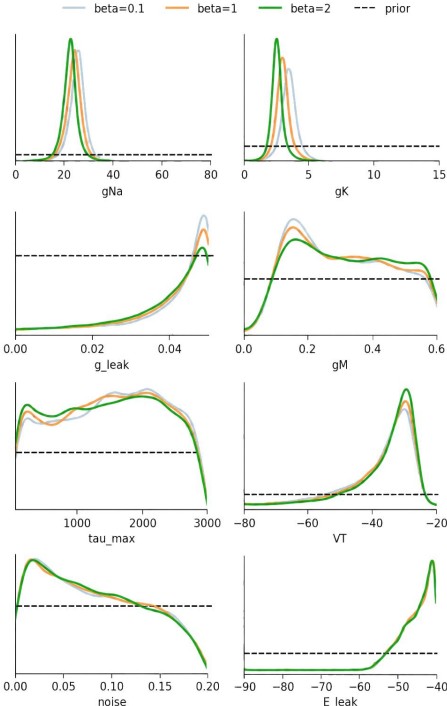

*Figure 2.* HH parameter marginals under `RouteB_NLE` for $\beta \in \{0.1, 1.0, 2.0\}$ (10K simulations).

cate that `RouteB_NLE` trained with 10K simulations yields stable posterior marginals across $\beta$ and produces posterior predictive simulations that match key qualitative features of the experimental voltage traces.

## 6. Route-specific trade-offs and limitations

Like other amortized inference methods, our approach trades substantial offline simulation and training cost for fast reuse at test time. When simulations are prohibitively expensive or the data-generating process shifts after training, per-instance adaptive inference may be more cost-effective and robust (Gutmann & Corander, 2016). Thus, our method is best viewed as complementary to traditional adaptive inference rather than a replacement.

**Route A.** Route A depends critically on the accuracy and stability of the learned joint score. Estimation errors can propagate along the annealing path and become amplified at large inverse temperatures, where the target posterior is more concentrated. In addition, posterior sampling relies on short-run Langevin or related MCMC updates, whose performance is sensitive to discretization steps, noise schedules, and the number of iterations. These choices require careful tuning and may be unreliable for sharp, multimodal, or weakly connected posterior geometries, especially as $\beta$ increases.

At the same time, Route A offers a useful advantage: by ex-

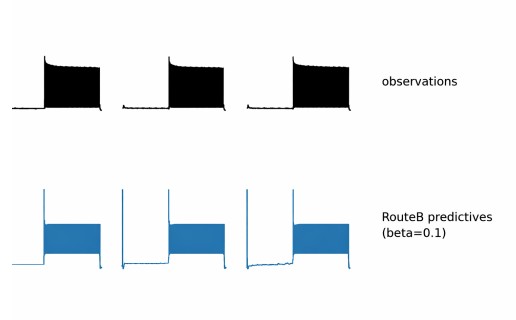

*Figure 3.* Three observations from the Allen Cell Types Database and `RouteB_NLE` predictive samples.

plicitly simulating from tempered joint distributions, it can generate samples that move off the nominal simulator manifold. Route A performs Langevin updates directly in the ambient joint space by targeting the tempered joint $p_\beta(\theta, x)$, so the generated pairs are not constrained to remain exact forward-simulation pairs, which is what we mean by "off-manifold". This ability is only useful to the extent that the learned score extrapolates locally beyond the nominal simulator-supported region; it is not an unrestricted guarantee far off-manifold. The potential benefit of this flexibility is precisely in misspecified settings, where the nominal simulator may under-cover regions that become relevant under the tempered target.

**Route B.** Route B avoids explicit MCMC and can therefore be faster at deployment time. However, its accuracy hinges on the quality and variance of the importance weights $w_\beta \propto \hat{p}_\eta(x \mid \theta)^{\beta-1}$ (NLE-based) and $w_\beta \propto \hat{r}^{\beta-1}$ (NRE-based). Classifier miscalibration or noisy likelihood-ratio estimates can lead to high-variance weights and severe effective sample size (ESS) collapse, especially when $|\beta - 1|$ is large or $\beta \to 0$. In such regimes, posterior estimates become unstable despite nominal amortization. Furthermore, reweighting cannot recover posterior regions that are entirely absent from the base joint distribution. If the nominal simulator undercovers regions emphasized by the tempered posterior, Route B can remain biased even with a large sample size. In contrast to Route A, it therefore offers limited robustness to severe model misspecification. As a variance-control extension, Appendix B.1 gives a defensive SNIS proposal that mixes the simulator joint with the product of marginals and proves bounded defensive weights for $\beta \in [0, 1]$.

**Generalization.** Both routes require a single conditional model $q_\phi(\theta \mid x, \beta)$ to generalize jointly over observations and inverse temperatures. Performance can degrade for $\beta$ values outside the trained range $\mathcal{B}$ or for rare and distribution-shifted observations. Furthermore, NPE inher-

its sensitivity to neural architecture and training hyperparameters, and suboptimal choices can lead to underfitting, mode dropping, or miscalibration, necessitating careful tuning and diagnostic checks (Lueckmann et al., 2021; Hermans et al., 2020). In addition, for small $\beta$ the posterior is prior-dominated and inherits sensitivity to prior misspecification, consistent with observations in generalized Bayesian updating (Bissiri et al., 2016; Lyddon et al., 2019).

## 7. Conclusion and future work

We gave $x, \beta$-amortized posterior inference for generalized Bayesian inference via power posteriors, learning a single conditional model $q_\phi(\theta \mid x, \beta)$ that supports single-pass sampling at deployment for user-chosen temperatures. We instantiated two complementary training routes: (A) score-assisted synthesis of tempered pairs with offline distillation, and (B) sampler-free SNIS reweighting that reuses a base joint dataset across $\beta$. Across standard SBI benchmarks, the resulting amortized family closely tracks non-amortized tempered targets over a broad temperature range, while eliminating simulator calls and inference-time MCMC at test time. The remaining challenges are (i) coverage/weight stability as $|\beta-1|$ grows and (ii) offline cost when simulators are expensive.

**Future work.** A promising direction is to make the approach adaptive rather than fixed: use ESS/weight-dispersion to automatically select between Route B (when weights are stable) and Route A (when coverage is insufficient). Second, more robust reweighting can build on the defensive SNIS construction in Appendix B.1, together with calibrated ratio/likelihood estimators and sequential importance sampling (using $q_\phi(\theta \mid x, \beta)$ as a prior), to control heavy-tailed weights beyond the basic Route B regime. Finally, extend evaluation to real-world amortization settings (many datasets, shifting conditions) and study principled temperature selection as a robustness knob under misspecification.

In this paper we have presented theory and methods for the power posterior only. This is just one variant of GBI. However, our methods extend straightforwardly, at least to the distance-based losses considered in (Gao et al., 2023) (such as the energy score). Instead of estimating the amortised score function in Phase I of Route A we estimate the amortised loss, exactly as in (Gao et al., 2023), and then in Phase II we target the joint distribution on the right side of (1) using standard MCMC methods, but using the amortised loss estimate from Phase I in place of the exact loss, to get the necessary training samples. We then proceed as in Phase III of Route A.

## Impact Statement

This work enables amortized inference for generalized posteriors across tempering values $\beta$, reducing the cost of robustness and sensitivity analyses in simulation-based inference. By supporting fast posterior and predictive queries after training, it can improve accessibility and lower repeated sampling and compute overhead. Risks include misuse of amortized approximations without calibration, or misinterpreting tempered posteriors as fully Bayesian posteriors under misspecification, which may yield misleading uncertainty. We recommend reporting calibration and posterior-predictive diagnostics, monitoring importance-weight stability (e.g., ESS), and validating against higher-fidelity samplers on a small subset when feasible.

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

# Appendix

# A. Neural architectures

All methods that use score modeling follow Noise Conditional Score Networks (NCSN) (Song & Ermon, 2019) on noisy pairs and Langevin-type sampling for tempered joints (Vincent, 2011; **?**; Song & Ermon, 2019).

## A.1. Score-assisted route (Route A)

The implementation details of the three phases of Algorithm 2 follow.

**Phase I : Score network** $s_\psi(\theta, x, \sigma)$ We first fix the noise scales $\sigma_1 > \cdots > \sigma_T$ used for denoising score matching and for the annealed sampler in Phase II, together with the number of Langevin steps $\{K_j\}_{j=1}^T$ and the base step-size scale $\epsilon_\eta$. The score network concatenates $[e_\theta, e_x, e_\sigma]$ and maps the result through a residual MLP to produce a joint score.

- Parameter embedding $e_\theta$ - Input is $\theta$. MLP (width 128) with 3 residual blocks and LayerNorm after each block.
- Data embedding $e_x$ -Input is $x$. MLP (width 128) with 3 residual blocks and LayerNorm after each block.
- Noise-scale embedding $e_\sigma$ - Input is $\log \sigma$. 2-layer MLP (width 128) similar to time/noise embeddings used in diffusion models (Ho et al., 2020; Nichol & Dhariwal, 2021).

**Phase II : Synthesize tempered pairs** We initialize each chain from the base joint $(\theta, x) \sim \pi(\theta)p(x \mid \theta)$ and run annealed short-run Langevin dynamics using the same noise scales $\{\sigma_j\}_{j=1}^T$ fixed in Phase I. The sampler loops over the noise scales from large to small. At scale $\sigma_j$, we set

$$\eta_j = \epsilon_\eta \frac{\sigma_j^2}{\sigma_T^2},$$

and keep this step size fixed for all $K_j$ Langevin steps at that scale. Each step uses the tempered score $\beta\, s_\psi(\theta, x, \sigma_j) - (\beta - 1)(s_\pi(\theta), 0)$ before moving to the next noise scale. The base step-size scale $\epsilon_\eta$ controls the overall discretization: overly large values can introduce discretization bias or unstable moves, whereas overly small values can leave the short-run chain poorly mixed. We tune $\epsilon_\eta$ on a held-out diagnostic sweep and then use the resulting $\{\eta_j\}$ schedule across the corresponding Route A synthesis runs.

**Phase III : Posterior estimator** For each temperature, we train an SNPE posterior $q_\phi(\theta \mid x, \beta)$ using `sbi` with `density_estimator='nsf' or 'mdn'` (Greenberg et al., 2019; Bishop, 1994). Settings: batch 128, `stop_after_epochs=50`, validation fraction 0.1.

**Step-size ablation.** Because Route A relies on discretized short-run Langevin dynamics, the synthesis quality depends on the Langevin step-size schedule. We therefore ablate the base step-size scale $\epsilon_\eta$ on the Gaussian mixture benchmark at $\beta = 0.9$, using C2ST against the reference power posterior as the evaluation metric. Figure 4 shows a non-monotone dependence: large step sizes lead to poor agreement due to discretization error, while very small step sizes also degrade performance because the chains move too slowly within the fixed short-run budget. In this sweep, the lowest C2ST occurs at an intermediate step-size scale.

**Architectural notes.** All MLPs use residual connections (He et al., 2016) and LayerNorm (Ba et al., 2016). The design mirrors diffusion/score-modeling practice for conditioning on noise variables (Ho et al., 2020; Nichol & Dhariwal, 2021).

## A.2. SNIS-weighted route (Route B)

(i) *NLE*: learn a neural likelihood $\hat{p}_\eta(x \mid \theta)$ (Papamakarios et al., 2019); choose $m(x) = 1$ so the SNIS weight is $w_\beta(\theta, x) \propto \hat{p}_\eta(x \mid \theta)^{\beta-1}$.

(ii) *NRE*: learn a classifier $d_\psi(\theta, x)$ that discriminates joint vs. product-of-marginals (Hermans et al., 2020; Durkan et al., 2020; Thomas et al., 2022); set $\hat{r}_\psi = \frac{d_\psi}{1-d_\psi}$ and use $m(x) = p(x)^{1-\beta}$, giving $w_\beta(\theta, x) \propto \hat{r}_\psi(x \mid \theta)^{\beta-1}$.

**Inference.** For an observation $x_{\text{obs}}$ and temperature $\beta$, form the context $[z(x_{\text{obs}}), \beta]$ (with $z$ an identity or learned summary) and draw samples from $q_\phi(\theta \mid x_{\text{obs}}, \beta)$ in a single forward pass—no simulator or MCMC.

**Notes.** MDNs work well for low-dimensional multi-modal posteriors; in higher dimensions use flow-based $q_\phi$ (e.g., MAF/NSF) or other conditional estimators (Papamakarios et al., 2021).

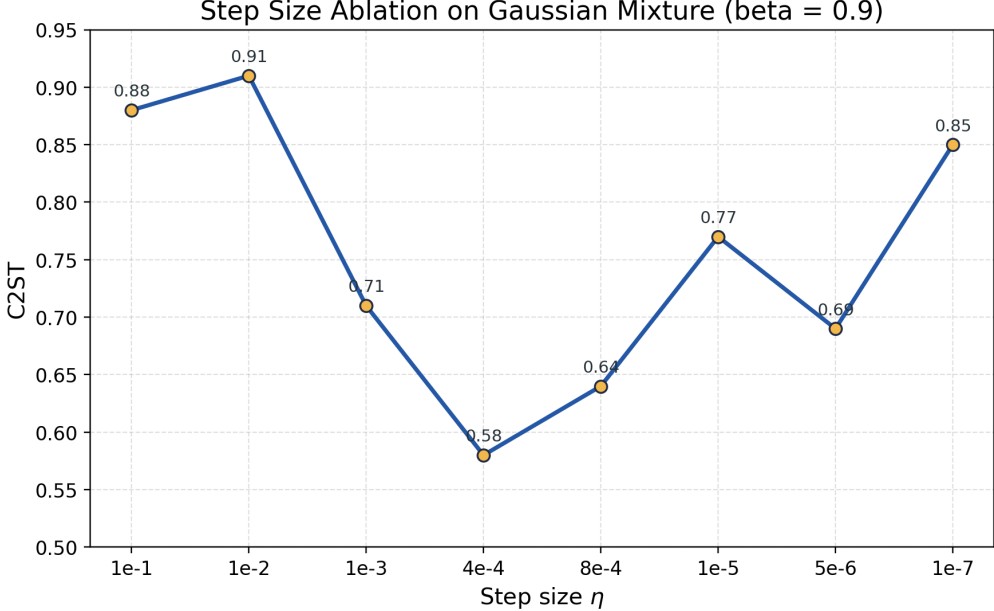

*Figure 4.* **Route A step-size ablation on Gaussian mixture.** C2ST between synthesized Route A samples and the reference power posterior at $\beta = 0.9$ for different Langevin step-size scales. Lower C2ST indicates closer agreement. The curve shows an intermediate optimum, with both overly aggressive and overly conservative Langevin updates producing worse samples.

## B. Proof of proposition

**Setup.** Let $\Theta \subseteq \mathbb{R}^{d_\theta}$ and $\mathcal{X} \subseteq \mathbb{R}^{d_x}$ be measurable spaces. Define the base joint distribution with density $p(\theta, x) = \pi(\theta)p(x \mid \theta)$. For $\beta > 0$, the tempered joint (unnormalized) is defined as $\tilde{p}_\beta(\theta, x) = \pi(\theta)p(x \mid \theta)^\beta m(x)$, where $m(x) > 0$ for any $x \in \mathcal{X}$. And its x-marginal $\tilde{p}_\beta(x) = \int \pi(\theta)p(x \mid \theta)^\beta \, d\theta \, m(x)$

Therefore, the conditional distribution can be written as

$$p_\beta(\theta \mid x) = \frac{\tilde{p}_\beta(\theta, x)}{\tilde{p}_\beta(x)} = \frac{\pi(\theta)p(x \mid \theta)^\beta}{\int \pi(\theta)p(x \mid \theta)^\beta \, d\theta} \propto \pi(\theta)p(x \mid \theta)^\beta. \tag{13}$$

which is usually the definition of the power posterior

$$p_\beta(\theta \mid x) = \frac{\pi(\theta)p(x \mid \theta)^\beta}{\tilde{p}_\beta(x)}. \tag{14}$$

Define the weights as:

$$w_\beta(\theta, x) = p(x \mid \theta)^{\beta - 1} m(x), \, m(x) > 0 \tag{15}$$

We train $q_\phi(\theta \mid x, \beta)$ by weighted conditional MLE on samples from $p(x, \theta)$:

$$\mathcal{L}(\phi) = \mathbb{E}_{(\theta, x) \sim p(\theta, x)}\big[\tilde{w}_\beta(\theta, x) \cdot (-\log q_\phi(\theta \mid x, \beta))\big], \qquad \tilde{w}_\beta(\theta, x) \propto w_\beta(\theta, x). \tag{16}$$

**Lemma B.1.** *For any measurable* $g : \Theta \times \mathcal{X} \to \mathbb{R}$,

$$\int g(\theta, x) \, p(\theta, x) \, p(x \mid \theta)^{\beta - 1} \, m(x) \, d\theta \, dx = \int g(\theta, x) \, \tilde{p}_\beta(\theta, x) \, d\theta \, dx \tag{17}$$

$$= \int \left( \int g(\theta, x) \, p_\beta(\theta \mid x) \, d\theta \right) \tilde{p}_\beta(x) dx. \tag{18}$$

*Proof.*

$$\int g(\theta, x)\, p(\theta, x)\, p(x \mid \theta)^{\beta - 1}\, m(x)\, d\theta\, dx = \int g(\theta, x)\, \pi(\theta)\, p(x \mid \theta)^{\beta}\, m(x)\, d\theta\, dx \tag{19}$$

$$= \int g(\theta, x)\, \tilde{p}_{\beta}(\theta, x)\, d\theta\, dx \tag{20}$$

where we used the decomposition of the base joint distribution and the definition of the tempered joint distribution, which proves (17). For (18), write $\tilde{p}_{\beta}(\theta, x) = p_{\beta}(\theta \mid x)\, \tilde{p}_{\beta}(x)$ and apply Fubini/Tonelli to exchange the order of integration:

$$\int g(\theta, x)\, \tilde{p}_{\beta}(\theta, x)\, d\theta\, dx = \int \Big( \int g(\theta, x)\, p_{\beta}(\theta \mid x)\, d\theta \Big)\, \tilde{p}_{\beta}(x)\, dx$$

$\square$

**Proposition 4.1:** Assume $\rho_{\beta}(x) > 0$ for $p_{\beta}$-almost all $x$ and that the model class for $q_{\phi}(\cdot \mid x, \beta)$ contains $p_{\beta}(\cdot \mid x)$. Then the population minimizers of (16) satisfy, for $p_{\beta}$-almost all $x$,

$$\arg\min_{\phi} \mathcal{L}(\phi) = \arg\min_{\phi} \int \rho_{\beta}(x)\, \mathrm{KL}\big(p_{\beta}(\cdot \mid x) \,\|\, q_{\phi}(\cdot \mid x, \beta)\big)\, dx. \tag{21}$$

*Proof.* By Lemma 1, up to a positive constant the objective equals

$$\mathcal{L}(\phi) \propto \int \left[ \int \big( -\log q_{\phi}(\theta \mid x, \beta)\big)\, p_{\beta}(\theta \mid x)\, d\theta \right] \tilde{p}_{\beta}(x)\, dx \tag{22}$$

$$= \int \Big( H\big(p_{\beta}(\cdot \mid x)\big) + \mathrm{KL}\big(p_{\beta}(\cdot \mid x) \,\|\, q_{\phi}(\cdot \mid x, \beta)\big) \Big)\, \tilde{p}_{\beta}(x)\, dx \tag{23}$$

$$= \int H\big(p_{\beta}(\cdot \mid x)\big)\, \tilde{p}_{\beta}(x)\, dx \; + \; \int \mathrm{KL}\big(p_{\beta}(\cdot \mid x) \,\|\, q_{\phi}(\cdot \mid x, \beta)\big)\, \tilde{p}_{\beta}(x)\, dx. \tag{24}$$

Here $H(\cdot)$ is the conditional entropy, independent of $\phi$. Since $\rho_{\beta}(x) > 0$, the integral is minimized iff $\mathrm{KL}\big(p_{\beta}(\cdot \mid x) \,\|\, q_{\phi}(\cdot \mid x, \beta)\big) = 0$ for $p_{\beta}$-a.e. $x$. This happens exactly when $q_{\phi}(\cdot \mid x, \beta) = p_{\beta}(\cdot \mid x)$ almost everywhere.

Then we have

$$\arg\min_{\phi} \mathcal{L}(\phi) = \arg\min_{\phi} \int \rho_{\beta}(x)\, \mathrm{KL}\big(p_{\beta}(\cdot \mid x) \,\|\, q_{\phi}(\cdot \mid x, \beta)\big)\, dx. \tag{25}$$

$\square$

**Proposition 4.2.** For $\beta \in [\frac{1}{2}, 1]$, with the NRE self-normalized IS weight

$$w_{\beta}(\theta, x) = p(x \mid \theta)^{\beta - 1}\, p(x)^{1 - \beta}.$$

Then

$$\mathbb{E}_{(\theta, x) \sim \pi(\theta) p(x \mid \theta)}[w_{\beta}^2] = \int\!\!\int_{\Theta}\!\!\int_{\mathcal{X}} \pi(\theta)\, p(x \mid \theta)^{2\beta - 1}\, p(x)^{2 - 2\beta}\, dx\, d\theta \; \le \; 1.$$

In particular, the SNIS estimator based on joint distribution $p(\theta, x)$ and $w_{\beta}$ has finite variance.

*Proof.* Fix $\beta \in [\frac{1}{2}, 1]$. For $\beta = \frac{1}{2}$ we have

$$\int_{\mathcal{X}} p(x \mid \theta)^{0}\, p(x)^{1}\, dx = \int_{\mathcal{X}} p(x)\, dx = 1,$$

hence $\mathbb{E}_{p(\theta, x)}[w_{\frac{1}{2}}^2] = 1$.

For $\frac{1}{2} < \beta < 1$, set

$$u = \frac{1}{2\beta - 1}, \qquad v = \frac{1}{2 - 2\beta}, \qquad \frac{1}{u} + \frac{1}{v} = 1, \quad u, v > 1.$$

By Hölder's inequality,

$$\int_{\mathcal{X}} p(x \mid \theta)^{2\beta - 1} p(x)^{2 - 2\beta} \, dx = \int_{\mathcal{X}} \left( p(x \mid \theta) \right)^{1/u} \left( p(x) \right)^{1/v} dx \le \left( \int p(x \mid \theta) dx \right)^{1/u} \left( \int p(x) \, dx \right)^{1/v} = 1.$$

Integrating the above bound against $\pi(\theta) \, d\theta$ yields $\mathbb{E}_{p(\theta,x)}[w_\beta^2] \le 1$. Finite variance follows since $Var[w_\beta] = \mathbb{E}_{p(\theta,x)}[w_\beta^2] - E_{p(\theta,x)}[w_\beta]^2 < E_{p(\theta,x)}[w_\beta^2] < \infty$. $\qquad \square$

### B.1. Defensive SNIS for Route B

The finite-variance guarantee in Proposition 4.2 applies to the original NRE-based Route B weights when $\beta \in [1/2, 1]$. A natural extension is to use a defensive proposal on the joint space, mixing the simulator joint with the product of marginals:

$$p_\lambda(\theta, x) = (1 - \lambda) \, \pi(\theta) p(x \mid \theta) + \lambda \, \pi(\theta) p(x), \qquad \lambda \in (0, 1).$$

Samples from $p_\lambda$ are obtained by drawing a Bernoulli variable: with probability $1 - \lambda$ sample $(\theta, x) \sim \pi(\theta) p(x \mid \theta)$, and with probability $\lambda$ sample $\theta \sim \pi(\theta)$ and $x \sim p(x)$ independently. The latter is available in SBI by simulating $\theta' \sim \pi(\theta')$, $x \sim p(x \mid \theta')$, and discarding $\theta'$.

For the NRE-based target

$$\tilde{p}_\beta(\theta, x) = \pi(\theta) p(x \mid \theta)^\beta p(x)^{1-\beta},$$

write the likelihood ratio as

$$r(\theta, x) = \frac{p(x \mid \theta)}{p(x)}.$$

Then

$$\tilde{p}_\beta(\theta, x) = \pi(\theta) p(x) r(\theta, x)^\beta, \qquad p_\lambda(\theta, x) = \pi(\theta) p(x) \{(1 - \lambda) r(\theta, x) + \lambda\}.$$

The defensive importance weight is therefore

$$w_{\beta,\lambda}(\theta, x) \propto \frac{\tilde{p}_\beta(\theta, x)}{p_\lambda(\theta, x)} = \frac{r(\theta, x)^\beta}{(1 - \lambda) r(\theta, x) + \lambda}. \tag{26}$$

With a trained NRE ratio $\hat{r}_\psi(\theta, x)$, we compute these weights in the log domain:

$$\log \hat{w}_{\beta,\lambda,i} = \beta \log \hat{r}_\psi(\theta_i, x_i) - \log \big((1 - \lambda) \hat{r}_\psi(\theta_i, x_i) + \lambda\big),$$

and use global per-temperature normalizers

$$S_{\beta,\lambda} = \log \sum_{i=1}^{N} \exp(\log \hat{w}_{\beta,\lambda,i}), \qquad \tilde{w}_{\beta,i}^{(\lambda)} = \exp(\log \hat{w}_{\beta,\lambda,i} - S_{\beta,\lambda}).$$

As $\lambda \downarrow 0$, the proposal reduces to the simulator joint and (26) recovers the original NRE weight $w_\beta(\theta, x) \propto r(\theta, x)^{\beta-1}$.

**Proposition B.2.** *Let $r(\theta, x) = p(x \mid \theta)/p(x)$, fix $\lambda \in (0, 1)$ and $\beta \in [0, 1]$, and define*

$$W_{\beta,\lambda}(r) = \frac{r^\beta}{(1 - \lambda) r + \lambda}, \qquad r > 0.$$

*Under the defensive proposal $p_\lambda$, the unnormalized defensive SNIS weight $w_{\beta,\lambda}(\theta, x) = W_{\beta,\lambda}(r(\theta, x))$ is bounded. Consequently,*

$$\mathrm{Var}_{p_\lambda}\big[w_{\beta,\lambda}(\theta, x)\big] < \infty.$$

*Proof.* For $\beta = 0$,

$$W_{0,\lambda}(r) = \frac{1}{(1 - \lambda)r + \lambda} \leq \frac{1}{\lambda}.$$

For $\beta = 1$,

$$W_{1,\lambda}(r) = \frac{r}{(1 - \lambda)r + \lambda} \leq \frac{1}{1 - \lambda}.$$

It remains to consider $\beta \in (0, 1)$. Let

$$F(r) = \log W_{\beta,\lambda}(r) = \beta \log r - \log((1 - \lambda)r + \lambda).$$

Then

$$F'(r) = \frac{\beta}{r} - \frac{1 - \lambda}{(1 - \lambda)r + \lambda} = \frac{(\beta - 1)(1 - \lambda)r + \beta\lambda}{r((1 - \lambda)r + \lambda)}.$$

Thus $F$ increases on $\big(0, \beta\lambda/((1 - \beta)(1 - \lambda))\big)$ and decreases afterwards. The maximum occurs at

$$r^{\star} = \frac{\beta\lambda}{(1 - \beta)(1 - \lambda)}.$$

Evaluating $W_{\beta,\lambda}$ at $r^{\star}$ gives the finite bound

$$W_{\beta,\lambda}(r) \leq \frac{\beta^{\beta}(1 - \beta)^{1-\beta}}{\lambda^{1-\beta}(1 - \lambda)^{\beta}} =: C_{\beta,\lambda} < \infty.$$

Combining the three cases, $w_{\beta,\lambda}$ is bounded for all $\beta \in [0, 1]$ by some finite constant $C'_{\beta,\lambda}$. Hence

$$\mathbb{E}_{p_\lambda}[w^2_{\beta,\lambda}] \leq (C'_{\beta,\lambda})^2 < \infty,$$

and the variance is finite. $\qquad\square$

*Remark* B.3. For $\beta > 1$, $W_{\beta,\lambda}(r) \sim r^{\beta-1}/(1 - \lambda)$ as $r \to \infty$, so boundedness no longer holds. Finite variance then requires additional moment conditions on the likelihood ratio under $p_\lambda$.

**Proposition B.4.** *Fix a temperature $\beta$ and let $\{\ell_i(\phi)\}_{i=1}^N$ be differentiable per-sample losses with*

$$\ell_i(\phi) = -\log q_\phi(\theta_i \mid x_i, \beta),$$

*Define the globally and locally normalized SNIS weights*

$$\tilde{w}^G_{\beta,i} = \frac{w_{\beta,i}}{\sum_{j=1}^N w_{\beta,j}}, \qquad \tilde{w}^L_{\beta,i} = \frac{w_{\beta,i}}{\sum_{j \in \mathcal{I}} w_{\beta,j}}.$$

*Let $\mathcal{I} \subset \{1, \ldots, N\}$ be a mini-batch of size $|\mathcal{I}| = b$. Define the global and local estimators*

$$\widehat{\mathcal{L}}^G_{\mathcal{I}}(\phi) = \frac{N}{b} \sum_{i \in \mathcal{I}} \tilde{w}^G_{\beta,i} \ell_i(\phi), \quad \widehat{\mathcal{L}}^L_{\mathcal{I}}(\phi) = \frac{N}{b} \sum_{i \in \mathcal{I}} \tilde{w}^L_{\beta,i} \ell_i(\phi)$$

*Then*

$$\mathbb{E}_{\mathcal{I}}\Big[\widehat{\mathcal{L}}^G_{\mathcal{I}}(\phi)\Big] = \sum_{i=1}^N \tilde{w}^G_{\beta,i} \ell_i(\phi), \quad \mathbb{E}_{\mathcal{I}}\Big[\widehat{\mathcal{L}}^L_{\mathcal{I}}(\phi)\Big] \neq \sum_{i=1}^N \tilde{w}^G_{\beta,i} \ell_i(\phi)$$

*Proof.* For each $i$, let $\mathbf{1}\{i \in \mathcal{I}\}$ be the inclusion indicator. If $\mathcal{I}$ is a uniform size-$b$ subset (with or without replacement), then $\mathbb{P}(i \in \mathcal{I}) = b/N$. By linearity of expectation and of the gradient operator,

$$\mathbb{E}_{\mathcal{I}}\Big[\widehat{\mathcal{L}}^G_{\mathcal{I}}(\phi)\Big] = \frac{N}{b} \sum_{i=1}^N \tilde{w}^G_{\beta,i} \mathbb{E}_{\mathcal{I}}[\mathbf{1}\{i \in \mathcal{I}\}] \ell_i(\phi) = \sum_{i=1}^N \tilde{w}^G_{\beta,i} \ell_i(\phi).$$

In contrast,

$$\mathbb{E}_{\mathcal{I}}\left[\widehat{\mathcal{L}}_{\mathcal{I}}^{(L)}(\phi)\right] = \sum_{i=1}^{N} \alpha_i \, \ell_i(\phi), \quad \text{with} \quad \alpha_i := \frac{N}{b} \, \mathbb{E}_{\mathcal{I}}\left[\frac{w_{\beta,i} \, \mathbf{1}\{i \in \mathcal{I}\}}{\sum_{j \in \mathcal{I}} w_{\beta,j}}\right].$$

To be unbiased for all losses $\{\ell_i\}$, we would need $\alpha_i = \tilde{w}_{\beta,i}^{G}$ for every $i$. However, this fails in general. A simple counterexample is $b = 1$: then $\sum_{j \in \mathcal{I}} w_{\beta,j} = w_{\beta,I}$ for the unique index $I \in \{1, \ldots, N\}$, and

$$\alpha_i = N \, \mathbb{E}_{\mathcal{I}}\left[\frac{w_{\beta,i} \, \mathbf{1}\{i = I\}}{w_{\beta,I}}\right] = N \, \frac{1}{N} \cdot 1 = 1,$$

so

$$\mathbb{E}_{\mathcal{I}}\left[\widehat{\mathcal{L}}_{\mathcal{I}}^{(L)}(\phi)\right] = \sum_{i=1}^{N} \ell_i(\phi),$$

whereas $\widehat{\mathcal{L}}_{\phi}(\beta) = \sum_{i=1}^{N} \tilde{w}_{\beta,i}^{G} \, \ell_i(\phi)$ with $\sum_i \tilde{w}_{\beta,i}^{G} = 1$ and $\tilde{w}_{\beta,i}^{G} \neq 1$ unless all $w_{\beta,i}$ are equal. Hence, the locally normalized estimator is biased whenever the weights are non-constant. $\square$

## C. Benchmark simulators and settings

### C.1. Two Moons

**Prior** $\qquad \mathcal{U}(-1,1)$

**Simulator** $\qquad x \mid \theta = \begin{bmatrix} r\cos\alpha + 0.25 \\ r\sin\alpha \end{bmatrix} + \begin{bmatrix} -|\theta_1 + \theta_2|/\sqrt{2} \\ (-\theta_1 + \theta_2)/\sqrt{2} \end{bmatrix}, \alpha \sim \mathcal{U}\left(-\frac{\pi}{2}, \frac{\pi}{2}\right), \ r \sim \mathcal{N}(0.1, 0.01^2)$

**Dimensionality** $\qquad \theta \in \mathbb{R}^2, \ x \in \mathbb{R}^2$

**References** $\qquad$ (Greenberg et al., 2019; Lueckmann et al., 2021)

### C.2. Gaussian Mixture

**Prior** $\qquad \mathcal{U}(-1,1)$

**Simulator** $\qquad x \mid \theta \sim \frac{1}{2}\mathcal{N}(\theta, I_2) + \frac{1}{2}\mathcal{N}(\theta, 0.01\, I_2)$

**Dimensionality** $\qquad \theta \in \mathbb{R}^2, \ x \in \mathbb{R}^2$

**References** $\qquad$ (Lueckmann et al., 2021; Sisson et al., 2007; Beaumont et al., 2009; Toni et al., 2009; Simola et al., 2021)

### C.3. SLCP

**Prior** $\qquad \mathcal{U}(-3,3)$

**Simulator** $\qquad x \mid \theta = (x_1, \ldots, x_4), \ x_i \sim \mathcal{N}(\mathbf{m}_\theta, \mathbf{S}_\theta),$

$\qquad\qquad \mathbf{m}_\theta = \begin{bmatrix} \theta_1 \\ \theta_2 \end{bmatrix}, \quad \mathbf{S}_\theta = \begin{bmatrix} s_1^2 & \rho s_1 s_2 \\ \rho s_1 s_2 & s_2^2 \end{bmatrix}, \quad s_1 = \theta_3^2, \ s_2 = \theta_4^2, \ \rho = \tanh\theta_5.$

**Dimensionality** $\qquad \theta \in \mathbb{R}^5, \quad x \in \mathbb{R}^8$

**References** $\qquad$ (Lueckmann et al., 2021; Papamakarios et al., 2019; Greenberg et al., 2019; Hermans et al., 2020; Durkan et al., 2020)

## C.4. Lorenz–96

**Prior**     $b_0 \sim \mathcal{U}[1.4, 2.2], \quad b_1 \sim \mathcal{U}[0.1, 1.0], \quad \sigma_e \sim \mathcal{U}[1.5, 2.5]$

**Simulator**     $x \mid \theta = (x_0, \dots, x_T), \ x_t \in \mathbb{R}^K$, obtained by integrating

$$\frac{dx_k}{dt} = -x_{k-1}(x_{k-2} - x_{k+1}) - x_k + 10 - \big(b_0 + b_1 x_k + \sigma_e \eta_k(t)\big), \quad \eta_k(t) \sim \mathcal{N}(0, 1),$$

for $k = 1, \dots, K$, with cyclic indices.

**Dimensionality**     $\theta \in \mathbb{R}^3, \quad x \in \mathbb{R}^{(T+1) \times K}$

**Fixed parameters**     State dimension $K = 8$;   time step $\Delta t = 3/40$;   trajectory length $T = 20$.
Initial condition $x_0 = \mathbf{1}_K$ (all ones).

**References**     (Lorenz, 1996)

# D. Route B effective sample size diagnostics

For Route B, we monitor the normalized effective sample size (nESS) of the SNIS weights across temperatures. For each benchmark, method, and $\beta$, nESS is computed separately for each held-out task using $K = 2000$ importance samples, and we report the mean and interquartile range over 30 independent tasks. The diagnostic measures weight degeneracy rather than posterior accuracy, but it is useful for identifying regimes where simple reweighting is expected to be unstable.

# E. Ground-truth power posteriors.

For each benchmark and temperature $\beta \in \{0.1, 0.3, 0.5, 0.7, 0.9, 1.0, 1.1, 1.3, 1.5\}$, we generate high-quality samples from the power posterior $p_\beta(\theta \mid x) \propto \pi(\theta) \, p(x \mid \theta)^\beta$ as follows.

**Two Moons.** We use a tempered random-walk Metropolis–Hastings kernel inside the uniform prior box. Proposals are reflected at the box boundaries, and we apply a small support-projection step to ensure the simulator's valid sector (implemented via the condition $v_x > 0$ in our likelihood). During burn-in we adapt the per-chain step sizes with a Robbins–Monro schedule to target an acceptance rate of 0.3. We run two chains initialized on opposite sides of the fold, burn in for 10,000 iterations, thin by 2, and then keep 5,000 states per chain, yielding 10,000 power posterior draws for each $\beta$. All computations use double precision.

**Gaussian Mixture.** We employ an independent rejection sampler that is exact for any $\beta > 0$. The proposal is a $\beta$-scaled Gaussian-mixture envelope $\sum_{i=1}^{2} \tilde{w}_i(\beta) \mathcal{N}(\theta; x, \ \Sigma_i/\beta)$ with $\Sigma_1 = I$ and $\Sigma_2 = 0.01\,I$, and weights $\tilde{w}_i(\beta) \propto (0.5)^{\max(\beta, 1)} K_\beta(\Sigma_i)$ (constant factors cancel in the acceptance ratio). A candidate outside the prior box is rejected; otherwise it is accepted with probability proportional to $\big[\,0.5\,\mathcal{N}(\theta; x, I) + 0.5\,\mathcal{N}(\theta; x, 0.01I)\,\big]^\beta / G_\beta(\theta)$, where $G_\beta$ denotes the envelope density. This yields 10,000 i.i.d. draws per temperature.

**SLCP.** We use a parallel–tempering random–walk Metropolis–Hastings sampler on the 5-dimensional box prior $\theta \sim \mathcal{U}([-3, 3]^5)$. For each target temperature $\beta$ we construct a geometric ladder of $R = 200$ inverse temperatures between $\beta_{\min} = 0.01$ and $\beta$, ordered from hot to cold. Each replica runs a Gaussian random–walk within the prior box; proposals outside the box are rejected. The per–coordinate proposal scale at temperature $\beta_r$ is set to $0.2/\sqrt{\beta_r}$ and is further adapted during burn–in via a Robbins–Monro schedule to target an average acceptance rate of 0.25. Between swap attempts, each chain performs 5 local MH updates, and every two iterations we attempt swaps between neighboring temperatures using an even–odd scheme. The temperature ladder is initialized by drawing 2048 prior samples and selecting the $R$ highest–likelihood points as starting states. We run 15 000 iterations in total, discard the first 5 000 as burn–in, and then retain every cold–chain state, yielding 10 000 draws from the power posterior $p_\beta(\theta \mid x)$ for each $\beta$.

**Lorenz–96.** For the Lorenz–96 task we run a single–chain random–walk Metropolis–Hastings sampler on the 3-dimensional box prior $b_0 \sim \mathcal{U}[1.4, 2.2]$, $b_1 \sim \mathcal{U}[0.1, 1.0]$, $\sigma_e \sim \mathcal{U}[1.5, 2.5]$. Likelihoods are evaluated via precomputed sufficient statistics of the discretized Gaussian transition model, so that each evaluation is $\mathcal{O}(1)$ in the number of particles. For each $\beta$ we initialize the chain by drawing 1500 prior samples, computing their likelihoods, and starting from the maximum–likelihood point. Proposals are Gaussian random–walks with independent components scaled to one sixth of the corresponding prior

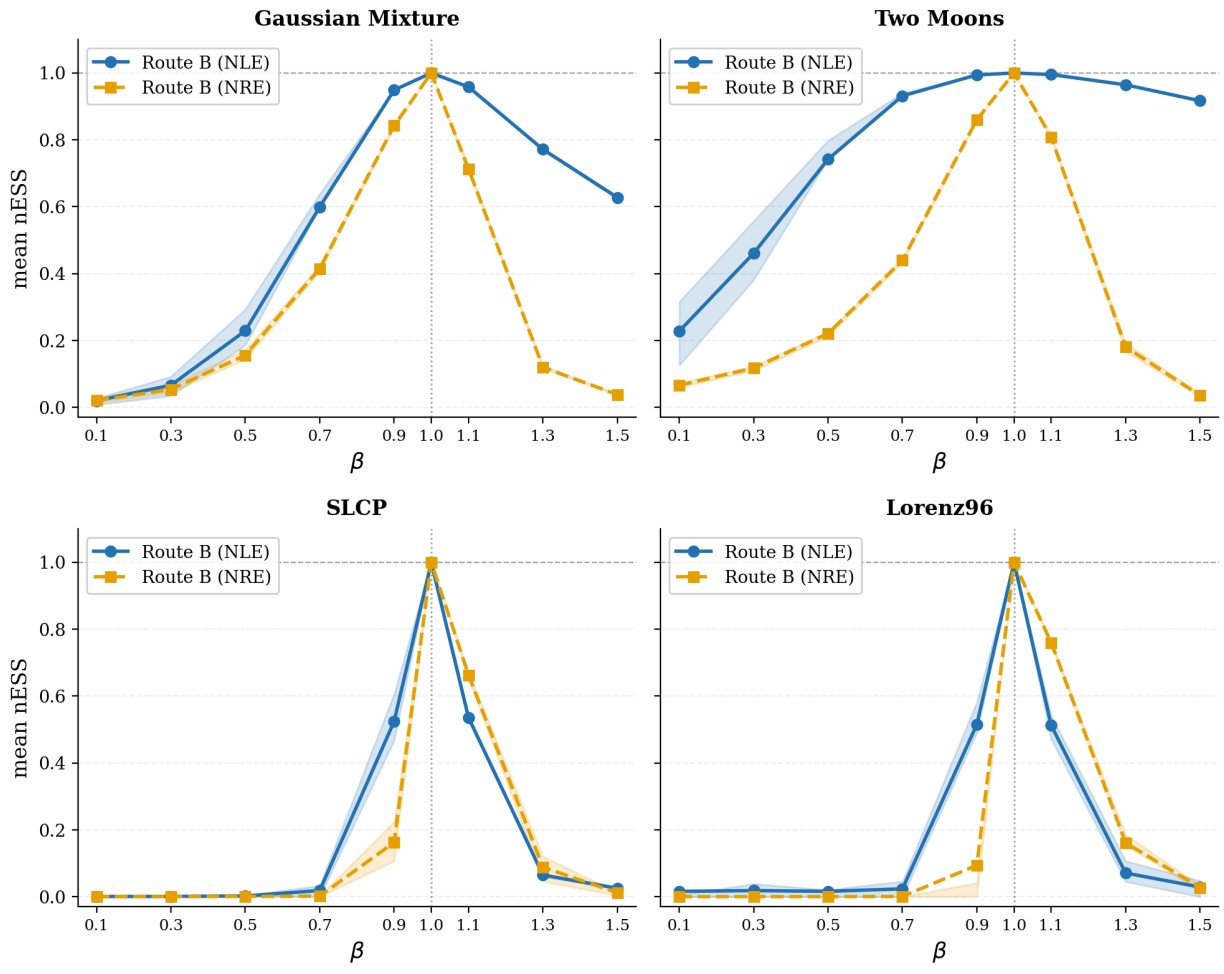

Shaded: 25–75th percentile across 30 independent joint draws ($K$=2000 samples/draw, $n_{train}$=10000)

*Figure 5.* **Route B effective-sample-size diagnostics.** Mean normalized ESS (nESS) for NLE- and NRE-based Route B weights across $\beta$ on the four benchmarks, with shaded regions denoting the 25–75th percentiles over 30 held-out tasks. The simulation budget is $n_{\text{train}} = 10{,}000$ and each nESS estimate uses $K = 2000$ importance samples. ESS is highest near $\beta = 1$, where the base joint proposal is closely matched to the target, and decreases as $\beta$ moves away from this regime, especially for SLCP and Lorenz–96.

width; they are reflected at the box boundaries to enforce the prior support. During a tuning phase of $6\,000$ iterations we adapt a single global step factor using a Robbins–Monro scheme to target an acceptance rate of $0.3$, with the acceptance probability based on the power posterior $p_\beta(\theta \mid x) \propto \pi(\theta)\, p(x \mid \theta)^\beta$ (the uniform prior cancels within the box). After tuning, we fix the step size and run $10\,000$ further MH iterations, collecting all states as reference draws from the power posterior for the given $\beta$.

## F. Hodgkin–Huxley model

We follow the single-compartment Hodgkin–Huxley (HH) model used in prior SBI work (Gonçalves et al., 2020; Pospischil et al., 2008). The model describes the membrane potential dynamics of a neuron under a fixed current injection protocol via standard conductance-based equations with voltage-dependent gating variables.

**Parameters.** The simulator involves eight unknown parameters, $\theta = (g_{\text{Na}}, g_{\text{K}}, g_{\text{M}}, g_{\text{leak}}, -V_T, -E_{\text{leak}}, t_{\text{max}}, \sigma) \in \mathbb{R}^8$, where $g_{\text{Na}}$, $g_{\text{K}}$, $g_{\text{M}}$, and $g_{\text{leak}}$ are maximal conductances for sodium, delayed-rectifier potassium, slow voltage-dependent potassium (M-type), and leak channels, respectively; $-V_T$ denotes an effective spiking threshold parameter; $-E_{\text{leak}}$ is the leak reversal potential; $t_{\text{max}}$ controls the time scale of the stimulus (or the recording window) in the benchmark

implementation; and $\sigma$ denotes the observation noise level. (All parameter bounds and priors are taken from the benchmark specification.)

**Observations and summary statistics.** Given $\theta$, the simulator produces a membrane-voltage trace $v(t)$, which we map to a 7-dimensional vector of summary statistics $x \in \mathbb{R}^7$. Specifically, we use the spike count, the mean resting potential, the standard deviation of the resting potential, and the first four central moments of the voltage trace (mean, standard deviation, skewness, and kurtosis), following (Gonçalves et al., 2020). These summary statistics constitute the observation used by all inference methods in the HH experiments.

# G. Additional visualizations: effect of $\beta$ on the posterior

To build intuition for how the power posterior changes with the tempering parameter $\beta$, we visualize samples from (i) a reference sampler and (ii) our two amortized routes on two toy problems (Gaussian mixture and two moons). In all panels, rows correspond to different $\beta$ values, and columns compare the reference power posterior with Route A and Route B. As $\beta$ decreases, the posterior becomes flatter and closer to the prior; as $\beta$ increases, it concentrates around data-fitting regions.

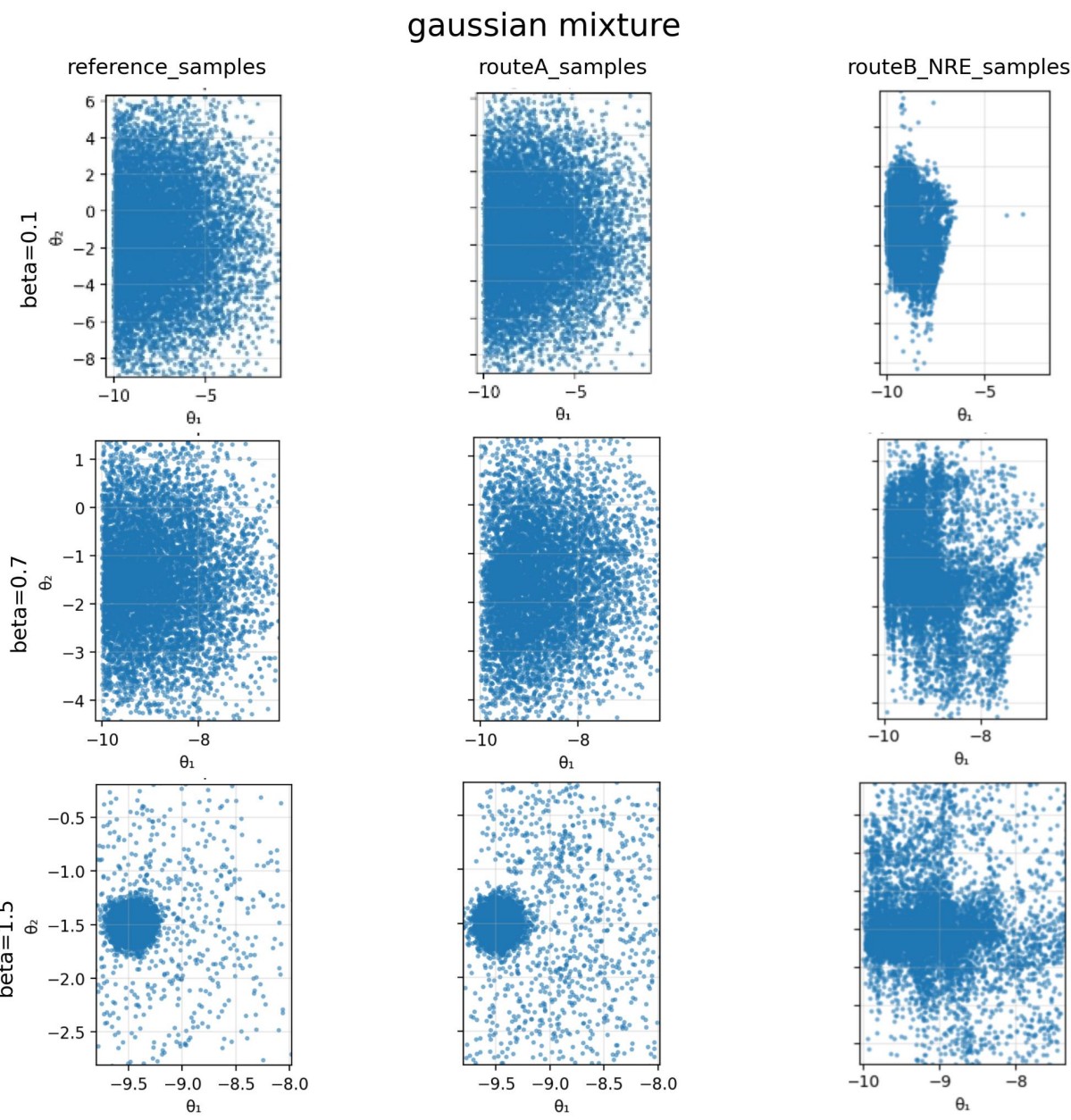

*Figure 6.* **Gaussian mixture.** Qualitative effect of the power posterior across different $\beta$ values. Rows correspond to $\beta \in \{0.1, 0.7, 1.5\}$ and columns show samples from the reference power posterior (left), Route A (middle), and Route B (NRE-SNIS; right).

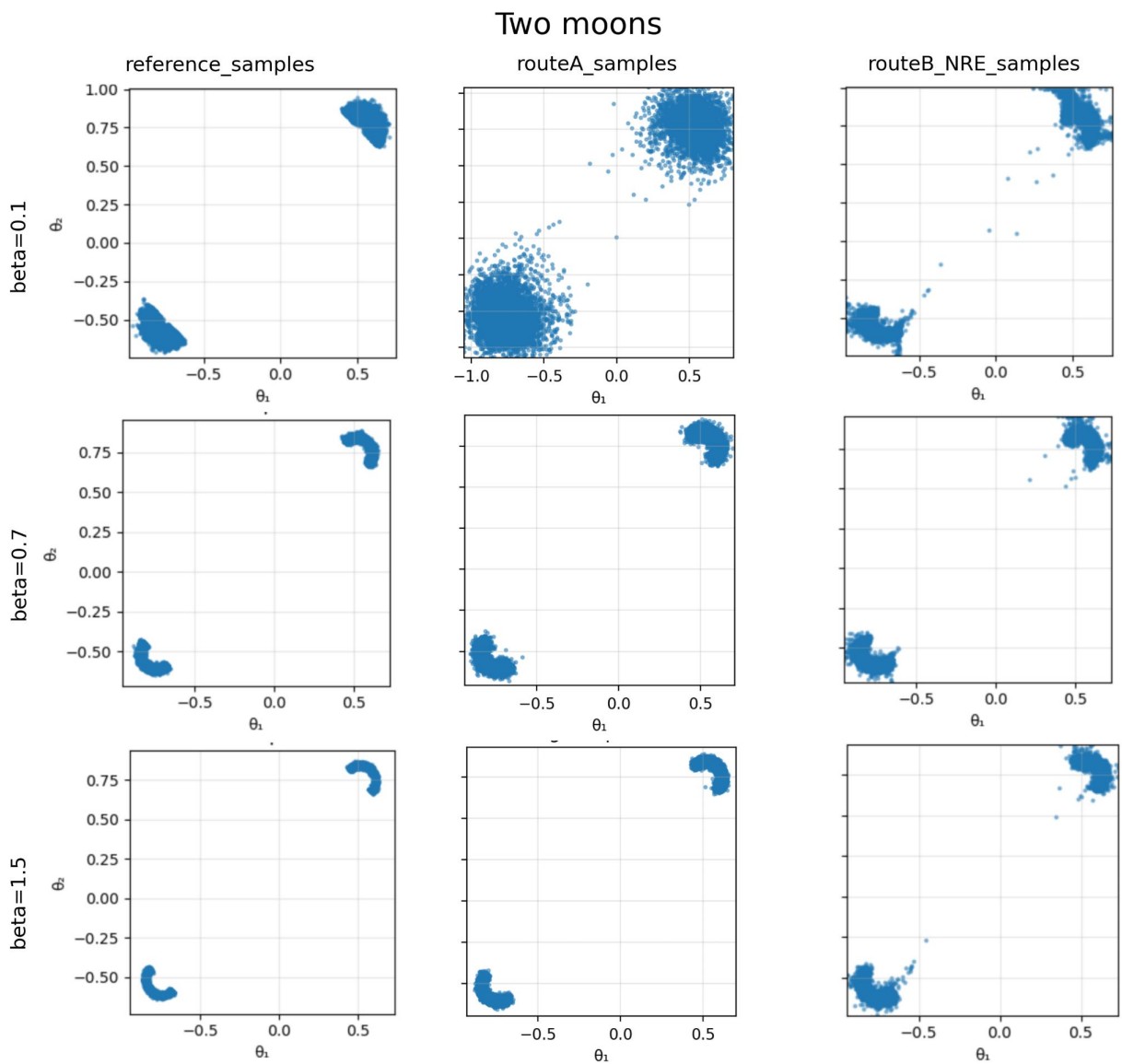

*Figure 7.* **Two moons.** Qualitative effect of the power posterior across different $\beta$ values. Rows correspond to $\beta \in \{0.1, 0.7, 1.5\}$ and columns show samples from the reference power posterior (left), Route A (middle), and Route B (NRE-SNIS; right).

