# OpenReview forum: "Amortized Simulation-Based Inference in Generalized Bayes via Neural Posterior Estimation"
_ICML.cc/2026/Conference — ICML 2026 regular_

### Official Review · Reviewer_RwN4 · 2026-03-01

**Soundness:** 2
**Presentation:** 3
**Significance:** 2
**Originality:** 3
**Overall Recommendation:** 4
**Confidence:** 4

**Summary:**

The paper provides two SBI methods targetting the power posterior (amortised across temperature).

In more detail, the power posterior modifies the ordinary Bayesian posterior by replacing the likelihood with the likelihood raised to a power $\beta$ (the temperature). It's often recommended as a way to deal with misspecification, and in recent years has also been interpreted as a type of generalized Bayesian method.

Simulation based inference approximates the power posterior even when the likelihood is intractable, based on samples from the model of interest.

**Compliance With Llm Reviewing Policy:**

Affirmed.

**Final Justification:**

Overall the paper seems like a useful methodological contribution, so I've increased my score from 3 (weak reject) to 4 (weak accept). As in the initial review I think soundness, presentation and originality are good. Some minor concerns about soundness and presentation have been addressed by the rebuttal. I think one could still argue about the strength of evidence in some of the results plots, but not to the extent of recommending rejection. For significance, the rebuttal has explained how the method would be used downstream, but for a higher score I'd have ideally liked an example showing that the method makes an important difference to an overall analysis.

**Key Questions For Authors:**

I think these are the most important questions from the weaknesses section, and addressing these would improve my score to weak accept.

1. In the paper's examples how would you choose $\beta$, and how would you use the results downstream?
2. Do the think the large C2ST values for the Lorenz96 model, and poor approximations in the top row of Figure 5 show that the methods sometimes work poorly? Or would you argue that this to be expected for these problems & the results are still useful in practice?
3. Can you report the weight variances or ESS values?

**Limitations:**

Yes

**Strengths And Weaknesses:**

I've highlighted the 4 criteria which the reviewer form mention in bold below

## Strengths

* The results seem **sound**. The methods have a good theoretical basis, with mathematical support. I checked most but not all parts of the proofs.
* Generally good **presentation**. The paper is well written and it's easy to follow the main ideas.
* The paper provides a sensible approach to an open problem of SBI for the power posterior amortising across $\beta$.
* The methods are **original**, building on previous work on related problems in a natural way, while making a novel contribution.

## Weaknesses (main)

* The title talks about "inference in generalized Bayes", but the paper is only about the particular case of a power posterior.
* The paper doesn't talk much about the **significance** of the power posterior. For instance, in the paper's examples how would you choose $\beta$, and how would you use the results downstream?
* I don't think the paper has a link to its code.
* There are some small issues with notation and minor mathematical details, which make parts of the paper slightly hard to follow and hinders **presentation**. See "Weaknesses (minor)" below.
* The joint score is a function of $\theta$ and $x$. So it could be hard to learn when either of these is high dimensional.
* Some of the results have quite large errors (reducing **soundness** of the results). For example Figure 1 shows all methods have large C2ST for the Lorenz96 model, and Figure 5 has poor approximations in the top row.
* Importance weight variances play an important role and are discussed in the theory and results. So it would be interesting to see estimates of them for the examples.
* The impact statement says "We recommend reporting calibration and posterior-predictive diagnostics, monitoring importance-weight stability (e.g., ESS)" But much of this is not reported in the paper's examples (only one posterior-predictive plot).

## Weaknesses (minor)

* Line 74 (right column) As written, I think the 2nd "=" in the display equation should be "$\propto$"
* Line 82 (right column) The objective has a sum over $\beta \in \mathcal{B}$, but at the end of the page it says $\mathcal{B}$ could be an interval.
* Line 132 (right column) "NCSN" isn't defined (I eventually noticed it's defined in Appendix A).
* Line 161 (left column) Where is the dependence on $\beta$ in the score? Is the idea that $t=\beta$?
* Line 254 (left column) I don't think you define $w_{\beta,i}$.
* Line 245 (right column) Earlier $\mathcal{B}$ was the set of $\beta$ values, so it might be good to use different notation for a batch.
* Figure 1: What is $n$ in the figure titles? In "route B", $N$ is the number of training samples. In other methods I'm not sure notation for number of training samples is explicitly introduced.

---

> ### Author Rebuttal · Authors · 2026-03-30
>
> 1. Scope: power posteriors vs generalized Bayes
>
> We agree that the current paper focuses on the power-posterior / beta-annealed likelihood family, and we will make this scope more explicit in the title and wording. This is a deliberate choice for the present paper, rather than a statement that the framework is inherently restricted to this setting. The same overall approach can in principle be extended to some other loss-based GBI objectives, as discussed in our response to Reviewer 2NcP, Q2. Nevertheless, our aim here is not to claim coverage of all possible GBI formulations, but to make the scope of the present paper explicit.
>
> 2. Significance of $\beta$ and downstream use
>
> We agree that the practical role of the power posterior should be explained more clearly. The choice of the tempering parameter is widely discussed in the generalized-Bayes literature. Our goal here is different: rather than proposing a new universal rule for selecting $\beta$, we make it computationally cheap to inspect the entire posterior family across $\beta$ once a single amortized network has been trained.
>
> This makes $\beta$ a practical robustness knob. After training $q_\phi(\theta\mid x,\beta)$, one can cheaply vary $\beta$ and examine how posterior summaries, posterior-predictive behavior, and stability diagnostics (e.g.\ ESS for RouteB) change across tempering levels, without rerunning expensive inference for every $(x,\beta)$ pair. In this sense, the main downstream use is robustness analysis and sensitivity assessment across $\beta$, rather than committing to a single universally optimal temperature. See also reviewer tbCp section 3 and 2NcP Q1.
>
> 3. Large C2ST values and difficult regimes
>
> We agree that some results correspond to genuinely difficult regimes. We do not interpret the large C2ST values in settings such as Lorenz--96 as evidence that the methods are uniformly poor. Rather, they reflect challenging regimes where the expected limitations become visible: for RouteB, importance-weight concentration and support mismatch; for RouteA, score-learning and discretization errors.
>
> We also note that C2ST can be quite sensitive. It is useful for comparing methods, but a moderately large C2ST value alone does not necessarily imply that the approximation is unusable. This is why we also report MMD as a complementary view of approximation quality. We will revise the discussion to make these points clearer and avoid over-claiming.
>
> 4. Minor comments and notation clarifications
>
> We thank the reviewer for the detailed minor comments. Some identify genuine presentation issues that we will fix, while others stem from notation that was not sufficiently explicit.
>
> In particular, we will clarify that the expression in the first comment is still unnormalized, so equality is intended there. We will also clarify that the experiments use a finite grid of $\beta$ values for convenience, although the method itself treats $\beta$ as a continuous conditioning variable. In addition, the learned score does not itself depend on $\beta$; $\beta$ only enters when forming the tempered target score. We will also make explicit that $w_{\beta,i}$ denotes the weight evaluated at the corresponding sample pair $(\theta_i,x_i)$.
>
> Finally, we agree that some notation should be distinguished more carefully. In particular, we will clarify that $n$ in Figure~1 denotes the simulation budget, i.e.\ the number of simulator calls / training samples.
>
> 5. ESS result
>
> We have also added an empirical ESS diagnostic as a function of $\beta$ for RouteB (anonymous figure link: \url{https://iili.io/BJrIvII.png}). For each task, proposal, and $\beta$, we compute the normalized ESS separately for each independent joint draw using its $K=2000$ importance samples, and then report the mean across draws; the shaded region shows the 25--75th percentile over the 30 draws. The results are consistent with the above interpretation: ESS is highest near $\beta=1$ and degrades as $\beta$ moves away from this regime, with substantially sharper degradation on the harder tasks (SLCP and Lorenz--96) than on the easier ones (Gaussian Mixture and Two Moons). This supports our interpretation that the practical deterioration of simple SNIS reweighting is driven by increasingly unstable importance weights, rather than by a failure of amortization itself. It also suggests that the severity of this effect depends on the underlying proposal quality, with the NLE-based proposal generally remaining more stable than the NRE-based one.
>
> 5. Reproducibility and code release
>
> We agree that releasing code would improve reproducibility. We will release the code upon acceptance, and in the revision we will also provide clearer implementation details to make the current experimental setup easier to reproduce.

---

> > ### Author Rebuttal · Reviewer_RwN4 · 2026-04-01
> >
> > Thanks for the clear answers to my queries and the extra work to plot the ESS results. Before giving a full reply, I wondered if you had any comment about the approximation quality in the top row of Figure 5? (part of my Q2)
> >
> > (Very minor point: I agree line 74 gives a valid importance weight. But using $=$ here doesn't seem consistent with $\propto$ on line 70. However this is not a substantive point and doesn't affect my score, so there's no need to discuss it further in this review process.)
> >
> > Edit: I wrote some final comments here, but then realised there's a "Final justification" box, so I moved them there instead.

---

> > > ### Author Response · Authors · 2026-04-01
> > >
> > > We agree that the top-row in Figure~5, namely the Two Moons example at $\beta=0.1$, looks visibly weaker for RouteA. Our point, however, is that Two Moons is a particularly special and challenging example for RouteA, despite its low dimension. The difficulty here is geometric rather than dimensional: posterior mass is concentrated near a narrow, strongly curved, near-manifold region because of the special likelihood, and the target density decays very rapidly away from it. This geometry becomes especially challenging at small $\beta$, where the tempered target is flatter and off-manifold effects from score-learning or Langevin discretization become more visible.
> > >
> > > Since RouteA explores in the joint $(\theta,x)$ space, it naturally produces some off-manifold exploration. In most examples, we regard this as a strength rather than a weakness, since it helps with support recovery and can be especially useful when the target posterior is not tightly confined to a thin manifold, or when tempering / proposal mismatch makes simple reweighting less reliable. For example, in the Gaussian Mixture task this exploratory behavior does not lead to the same visible distortion and instead remains compatible with good posterior coverage. In Two Moons, however, the geometry is much more restrictive, so the same behavior becomes especially visible rather than being averaged out over a thicker posterior support. We will clarify that this panel should therefore be interpreted as a particularly geometry-sensitive stress case, rather than as representative of the typical behavior across tasks.
> > >
> > > Thank you very much for the helpful follow-up, and also for taking the time to increase your score. I only saw this comment just now when checking the revision history, so apologies for the delayed reply.
> > >
> > > A concrete downstream use case is the misspecified setting in which one turns to generalized Bayes precisely because a standard Bayesian posterior may be too brittle. In such cases, the tempering level $\beta$ is typically not known and one often wants to inspect how the posterior changes across a range of $\beta$ values before deciding what level of robustness is appropriate. Without amortization, this would require rerunning inference for every new observation x and every candidate β, which is computationally expensive. In contrast, once our model $q_\phi(\theta|x,\beta)$ has been trained, the same amortized network can be reused for new observations and can immediately provide posterior samples across many $\beta$ values. This is the practical advantage of amortizing jointly over both x and $\beta$.

---

### Official Review · Reviewer_tbCp · 2026-03-10

**Soundness:** 3
**Presentation:** 3
**Significance:** 3
**Originality:** 3
**Overall Recommendation:** 3
**Confidence:** 3

**Summary:**

This paper considers amortized simulation-based inference for generalized Bayesian inference. Existing approaches typically require running MCMC or other sampling procedures separately for each observation $x$ and temperature $\beta$ which can be computationally expensive.

To solve this problem, this work proposes to learning a $(\beta, x)$-conditioned neural posterior estimator that amortizes inference across both the observation $x$ and the temperature parameter $\beta$. Two training strategies are proposed: (1) Score-based synthetic data generation; (2) Self-normalized importance sampling (SNIS) reweighting.

The paper also evaluated the performance across several experiments including standard SBI benchmarks and the single-compartment Hodgkin–Huxley model of neuronal voltage dynamics.

**Compliance With Llm Reviewing Policy:**

Affirmed.

**Final Justification:**

My concerns have been adequately addressed. The experiments are good but less convincing. Therefore I tend to keep score of 3.

**Key Questions For Authors:**

If it is possible for you to find real-world expensive scenarios to use your methods? Meanwhile, you should also release the code to ensure reproducibility.

**Limitations:**

Not applicable.

**Strengths And Weaknesses:**

+ Strengths:
   + The paper proposes  a novel and interesting idea that combines amortized learning, SBI and Generalized Bayes.
   + The method is well motivated. The paper is easy to follow and read.
   + Route B is more attractive as it doesnot require score matching and MCMC processes to optimize the joint approximate posterior $q$.

+ Weakness:
   + One main issue of this paper is: it actually only considers power posteriors, a simplified version of generalized Bayes.
   + The overall computational cost could be large due to the multi-phase training processes in Route A. This requires more discussions in the paper.
   + Lack of theoretical analysis.
   + Choice of $\beta$ is still not studied. Though it is not an issue for this paper.
   + Lack of expensive and persuasive SBI experiments. Or you should state why the experiment in Section 5.2 is a good illustration example for your methods, e.g. from the perspective of computational cost .
   + Lack of code as the supplementary material.

---

> ### Author Rebuttal · Authors · 2026-03-30
>
> 1. Scope and theoretical analysis
>
> We agree that the current paper focuses on the power-posterior / beta-annealed likelihood family, and we will make this scope more explicit in the title and wording. This is a deliberate scope choice for the present paper, rather than a statement that the framework is inherently restricted to this setting. The same overall approach can in principle be extended to some other loss-based GBI objectives, as discussed in our response to Reviewer 2NcP, Q2. Even with such extensions, however, the resulting framework would still not cover all possible GBI formulations, so our main concern here is to make the scope of the present paper explicit rather than to claim full generality.
>
> Within this scope, the paper provides theoretical analysis through Propositions-4.1 and 4.2: Proposition-4.1 shows that the SNIS-weighted objective targets the forward KL to the power posterior, and Proposition-4.2 identifies a finite-variance regime for the corresponding importance weights in RouteB. We agree, however, that the current theory does not cover all regimes considered in our experiments, and we will clarify these boundaries more explicitly in the revision. We also note that follow-up analysis with a defensive importance-sampling extension suggests a principled route to stabilizing the reweighting scheme beyond the basic setting; see also our response to Reviewer yt6F, Section~1.
>
> 2. Computational cost of Route A
>
> We agree that the computational cost of RouteA should be discussed more explicitly. RouteA is more flexible, but also more computationally expensive, due to score-based synthetic data generation in the joint $(\theta,x)$ space and the greater sensitivity of the Langevin procedure to its tuning parameters. This is why we view it as a complement to RouteB, rather than a universal default: RouteB is the practical choice when reweighting is stable, whereas RouteA is most useful when support recovery or off-manifold exploration is needed.
>
> We will make this trade-off clearer in the revision. The additional cost is primarily incurred during training, while test-time inference remains amortized; by contrast, existing generalized-Bayes baselines typically require separate sampling for each new $(x,\beta)$ pair.
>
> 3. Choice of $\beta$
>
> The selection of $\beta$ is itself an important topic in generalized Bayes. Our framework is compatible with many such approaches, as noted in our response to Reviewer 2NcP, Q1. Because it amortizes over both the data $x$ and the tempering level $\beta$, it supports fast posterior access across a range of $\beta$ values. This is useful, for example, in procedures that choose $\beta$ by optimizing criteria such as WAIC.
>
> 4. Experimental scope and the role of the Hodgkin--Huxley example
>
> We agree that the role of Section~5.2 should be clarified. The Hodgkin--Huxley example is not intended as a ground-truth posterior recovery benchmark like the synthetic SBI tasks. Rather, it serves as a realistic misspecified-data case study and as a first step beyond standard synthetic SBI benchmarks toward a more expensive scientific-simulator setting with model mismatch.
>
> From this perspective, its purpose is to illustrate a regime where amortizing across both datasets and tempering levels is practically meaningful, since repeated per-dataset, per-temperature generalized-Bayes inference would otherwise be comparatively costly. It also shows how the inferred posterior family changes with $\beta$ in a realistic misspecified setting, including cases where some marginals are relatively insensitive to tempering over the range shown. We will revise the text to make this motivation, including the computational perspective, more explicit.
>
> 5. Reproducibility and code release
>
> We agree that releasing code would improve reproducibility. We will release the code upon acceptance, and in the revision we will also provide implementation details to make the current experimental setup easier to reproduce.

---

> > ### Author Rebuttal · Reviewer_tbCp · 2026-04-02
> >
> > Thanks for the clarifications. I tend to keep my initial score. The experiments are good but not super convincing.

---

> > > ### Author Response · Authors · 2026-04-02
> > >
> > > Thank you again for the helpful feedback! We will make two points more explicit in the revision. For the Section 5.1 benchmarks, our claim is not that the amortized methods are uniformly best, but that they remain competitive with standard non-amortized SBI approaches while additionally amortizing over both observations and tempering levels. For Section 5.2, the Hodgkin--Huxley example is meant less as another ground-truth recovery benchmark and more as a first step toward a more realistic, computationally expensive, misspecified scientific-simulator setting, where this amortization is practically useful.

---

### Official Review · Reviewer_2NcP · 2026-03-12

**Soundness:** 3
**Presentation:** 3
**Significance:** 3
**Originality:** 3
**Overall Recommendation:** 4
**Confidence:** 3

**Summary:**

The paper presents three methods for generating training data for amortised posterior estimation in the setting of generalised Bayesian inference in which the likelihood term is tempered by variable inverse-temperature beta. The first method uses noise-based score estimation followed by Langevin dynamics for different beta (Route A). The second uses neural likelihood estimation followed by importance sampling with respect to the beta=1 case (Route B-NLE). The third replaces neural likelihood with neural likelihood ratio estimation (Route B-NRE). Each method requires a prior and the ability to sample a likelihood simulator, making this a simulation-based inference task. The three methods are applied to four synthetic benchmark SBI scenarios. Proof-of-principle is shown for the Route B-NLE in a complex Hodgkin-Huxley model with real data inference.

**Compliance With Llm Reviewing Policy:**

Affirmed.

**Final Justification:**

The topics of amortised inference and Generalised Bayesian Inference are of interest to the community. This includes the power-likelihood version implemented here. This work presents three sensible approaches and considers their relative strengths and weaknesses. The most valuable probing of the methods is on four synthetic SBI benchmarks in Fig 1. The authors have agreed to improve the interpretation of Fig 1, which shows the viability of the methods - I consider this to be important to give a clear and accurate picture of both successes and failures. The authors have clarified the role of the H-H experiment, which has some value, although it is relatively limited because it does not address the core research question of whether the method has learned an accurate amortisation of the $\beta$-posterior.

The work would be made stronger by (i) greater empirical characterisation of the success and failure modes, and (ii) comparison of the $\beta$-posterior against gold standard MCMC on real-world data beyond the SBI benchmarks. Some progress is made on point (i) by the provision of an ESS study in a parallel rebuttal.

Overall I recommend acceptance of this paper, as it would be useful for practitioners/researchers interested in amortised $\beta$-annealed inference.

**Soundness: 3**: The methodologies are well considered and solid.

**Originality: 3**: I am not aware of previous solutions to this problem.

**Significance: 3**: The problem is relevant to the field.

**Clarity: 3**: The paper is generally clear and well structured.

**Key Questions For Authors:**

1. For Route A, do you use samples from the same trajectory for different beta values? For Route B, do you use the same samples from the base distribution for different beta? If either case is yes, is there a risk that this would induce correlations that would break the i.i.d. assumption on training data and cause overfitting?
2. What do you see as the main challenges for amortising GBI with arbitrary loss functions? In principle Langevin dynamics (Route A) can sample from this as long as we can estimate its gradient wrt theta. Under what assumptions could we achieve this?
3. Am I correct in understanding that C2ST reports the classifier accuracy and so should be 0.5 for a perfectly described posterior?
4. What is fig2 supposed to communicate? Is there a reason these marginals should be the same for different beta?
5. Are the kernel used for MMD and classifier used for C2ST stated anywhere?

**Limitations:**

Yes

**Strengths And Weaknesses:**

Strengths:

- Full amortisation of generalised Bayesian inference when using a beta-annealed likelihood, without requiring expensive inference-time MCMC.
- All three of the presented approaches are well motivated and well thought through.
- Characteristic of SBI, the methods do not require analytically tractable likelihood or gradients, making them applicable to arbitrary simulators, which is crucial in many scientific domains.
- Methods tested on four synthetic SBI benchmarks of increasing complexity.
- Well written and structured paper, easy to follow.
- I appreciate the considered discussion about the methods' weaknesses as well as their strengths.

Ares for improvement:

- Experimental results in section 5.1:
   - The results in fig1 are somewhat difficult to read and interpret.
   - I don’t agree with many of the claims made about fig1. The relative performance is quite inconsistent between experiments. Route B is not always best around beta=1. The methods do not always closely match the ground-truth power posteriors or achieve near-chance C2ST, which I understand to mean a value of 0.5. The performance on some tasks can be quite poor on the more difficult benchmarks, e.g. Route B on SLCP. The amortised methods are not always comparable with the non-amortised ones, e.g. NLE on SLCP seems clearly best on C2ST.
   - It is not stated how large the datasets x are from which we calculate the posterior. Is it always a single datapoint, is it fixed to a larger number, or can it vary? This is important because larger datasets lead to more concentrated posteriors, especially for large beta, and I would expect this to impact the performance of different methods - e.g. importance weights for Route B might have rather high variance. It is also important for the choice of neural architecture. I think the “n” in fig1 is the number of datapoints used to train the network, not the number of samples per datapoint, since you claim it is related to data efficiency, which only makes sense in a training context.
  - Although some reasons are proposed, it would benefit the paper to provide evidence for why methods perform well/badly in different settings, as well as visualisations of x in addition to theta.
  - In general, it would be beneficial to attempt to optimise methods (within practicality).
- Experimental results in section 5.2:
   - The key experimental validation seems to be that samples from the posterior of Route B (NLE) induce x that look qualitatively similar to the data. This is a reasonable check but does not directly test the main research question - whether the three amortised variants are able to reconstruct accurate posteriors across a range of beta.
   - It’s not clear what we are supposed to conclude from fig2.

In summary, the paper presents a well motivated and analysed formulation. It would be improved by experimental evidence that (i) compares all three methods with correct-but-expensive MCMC, as in fig1, but for challenging real-world experimental scenarios akin to the setting of section 5.2; as well as more forensic studies into the sources of performance gains/drops, and optimisations. Finally, some care should be taken not to over claim on the performance of fig1.

Other notes:
- I would prefer that the title be more clear that you do not amortise all GBI, only the power-likelihood version. This is important because a reader might misunderstand that they can provide any GBI loss and have it amortised.
- Make sure beta is defined as the inverse temperature instead of temperature in all locations, since increasing temperature should increase entropy.

---

> ### Author Rebuttal · Authors · 2026-03-30
>
> 1. Interpretation of Figure~1 and clarification of the experimental setup
>
> We use "observation" in the paper (or “dataset” in the posterior-conditioning sense) for the d-dimensional input $x$, and “training data” for the collection $(\theta,x_i)_{i=1}^N$.
>
> We agree that we should explain Figure-1 more carefully. Each posterior in Figure-1 is conditioned on a single dataset, and the reported curves are then averaged over multiple held-out datasets, so the trends are not driven by any single realisation. We do not claim that the amortized methods are uniformly best across all benchmarks or all $\beta$ values. Rather, Figure~1 is intended to show that RouteA and RouteB are competitive with existing non-amortised SBI methods while additionally amortizing over both datasets and $\beta$. We will moderate any overly strong wording in discussion accordingly.
>
> More generally, we will clarify the full experimental setup in the revision, including how each evaluation posterior is constructed and what $n$ denotes in Figure-1, and the relevant architectural and evaluation details. In particular, in Figure-1, $n$ denotes the simulation budget, i.e. the number of simulator calls / training samples used to train the network (we apologise for the typo: here $n$ is the same quantity as $N$ in Algorithms~2 and 3). We will also state the metric definitions and implementations more explicitly: for C2ST, values near $0.5$ indicate near-indistinguishability between approximate and reference posterior samples; and for both the C2ST classifier and the MMD kernel, we follow standard choices from prior SBI work and will cite them explicitly.
>
> 2. Evidence for why methods are good/bad and optimization
>
> We agree that this mechanism-level discussion should be made more explicit. Section-6 already outlines some of these interpretations in connection with Figure-1, but we will revise it to diagnose the main sources of success and failure more directly. For example, for RouteA, we will add the requested step-size ablation (see reviewer yt6F, Section~3) to show how performance degrades as the Langevin step size moves away from a well-tuned value. The purpose of this ablation is not to propose a practical step-size selection rule for real applications, but rather to make the source of the breakdown more transparent retrospectively.
>
> So far, our optimisation efforts have mainly focused on hyperparameter selection; further refinements remain future work. See also our response to reviewer yt6F, Section~1.
>
> 3. Role of the Hodgkin--Huxley experiment and interpretation of Figure~2
>
> We agree that the role of Section-5.2 and Figure-2 should be stated more clearly. The H-H experiment is not intended as a ground-truth posterior recovery benchmark like the synthetic tasks in Section~5.1. Rather, it serves as a realistic misspecified-data case study, showing that the amortized power-posterior framework can still produce posterior samples with qualitatively plausible predictive traces, while allowing one to inspect how inference changes with $\beta$.
>
> Figure~2 is therefore not meant to suggest that posterior marginals should be identical across different $\beta$ values. Its purpose is to illustrate how the inferred posterior family varies across tempering levels in a realistic setting. In this example, some parameter marginals appear relatively insensitive to $\beta$ over the range shown. Our interpretation is that, for these directions, the data are already sufficiently informative that moderate changes in tempering do not substantially alter the posterior. Correspondingly, more pronounced changes only appear when $\beta$ is taken to a much larger or much smaller regime. We will extend the figure to a wider range of $\beta$ values and revise the discussion to make these points explicit.
>
> Q1: Shared samples across $\beta$.
> For RouteA, different $\beta$ values are independent and parameterise independent trajectories, run in parallel; we keep only the terminal sample from each trajectory. For RouteB, samples come from the same base distribution, but different $\beta$ values induce different importance weights and hence different weighted objectives. We will clarify this in the revision. Amortising over $\beta$ is important for downstream use. Many GBI methods for estimating $\beta$ rely on fast posterior sampling at many different $\beta$-values, and our framework supports this. See also reviewer tbCp section 2.5 and RwN4 section 2.
>
> Q2: Arbitrary-loss GBI.
> We agree this is important. Losses such as the energy score could in principle be incorporated into RouteA through a learned amortised loss/energy and standard MCMC on the resulting joint target. The same loss-estimate can be used in the weights in RouteB, though a well-adapted proposal and theory controlling weight variance would be needed. We will clarify this scope distinction in the revision.
>
> Our responses to Questions 3, 4, and 5 are given in Sections~1, 3, and 1, respectively.

---

> > ### Author Rebuttal · Reviewer_2NcP · 2026-04-03
> >
> > Thank you for your responses. Reading the rebuttals to other reviewers, I appreciate the ESS study which I believe adds value, as well as the definition of a defensive IS strategy.
> >
> > Just a couple of short comments/questions:
> >
> > 3. This is clear, thank you for explaining. Another explanation for the consistent marginals would be that $x$ contains little information about that parameter, so the posterior is prior-dominated, making it insensitive to changes in $\beta$. Perhaps this would be illuminated by adding a curve for the prior (also nicely serving as the $\beta\rightarrow0$ limit). It may also be the case that $x$ is informative about the joint structure but not the marginals. In any case, I leave these to your discretion, as they are not important for my evaluation or your rebuttal.
> >
> > Q6: On page 8 you claim that Route A can generate samples that move off the nominal simulator manifold. I would expect that, since the score model is trained on samples from $p_1(\theta,x)=\pi(\theta)p(x|\theta)$ [Algorithm 2], it would only be valid on this manifold of $(\theta,x)$ (modulo some uncontrolled OOD inductive bias). What am I missing here? Please can you clarify precisely which manifold you are referring to, and the extent and limit of the ability of Route A to leave it?

---

> > > ### Author Response · Authors · 2026-04-03
> > >
> > > Thank you, this is very helpful!
> > >
> > > For your point 3: We agree that another plausible explanation for the relatively stable marginals is that the data are only weakly informative about those parameters, so the corresponding posteriors remain close to prior-dominated over the β-range shown. We will mention this interpretation in the revision. We agree that adding the prior / $\beta$→0 limit would also help, if space permits.
> > >
> > > For Q6: Thank you for pushing us to be more precise here. What we meant by “off the nominal simulator manifold” was not that Route A can be trusted arbitrarily far outside the support on which the joint score is learned. Rather, the point is that Route A performs Langevin updates directly in the ambient joint $(\theta,x)$ space targeting the tempered joint $p_\beta(\theta,x) ∝ \pi(\theta)p(x|\theta)^\beta$, so the generated pairs are not constrained to remain exact forward-simulation pairs obtained by ancestral sampling from $\pi(\theta)p(x|\theta)$. In that sense, it can move away from the set of simulator-consistent pairs, which is what we intended by “off-manifold”. But this ability is only meaningful to the extent that the learned score extrapolates locally beyond the nominal simulator-supported region; it is not an unrestricted guarantee far into OOD regions. The potential benefit of this flexibility is precisely in misspecified settings, where the nominal simulator may under-cover regions that become relevant under the tempered target. We will revise the wording to make this distinction much clearer.

---

### Official Review · Reviewer_yt6F · 2026-03-14

**Soundness:** 3
**Presentation:** 3
**Significance:** 3
**Originality:** 3
**Overall Recommendation:** 4
**Confidence:** 4

**Summary:**

This paper amortizes inference over the family of power posteriors $p_\beta(\theta|x) \propto \pi(\theta)p(x|\theta)^\beta$ by training a single parameterized neural posterior estimator $q_\phi(\theta|x,\beta)$ conditioned on both observation and temperature. At test time, sampling from any power posterior reduces to a single forward pass with no MCMC or simulator calls. The approach is evaluated on standard SBI benchmarks.

**Compliance With Llm Reviewing Policy:**

Affirmed.

**Final Justification:**

The paper addresses a well-motivated problem of amortizing power-posterior inference over both observations and temperatures. The rebuttal adequately addressed my concerns (ESS diagnostics for β>1, clarification of Route A sensitivity), and I maintain my positive score.

**Key Questions For Authors:**

1. Can you extend the weight variance analysis (Proposition 4.2) to $\beta > 1$, or provide empirical ESS diagnostics as a function of $\beta$ for Route B? The current theory covers $\beta \in [1/2, 1]$ but experiments go to $\beta = 1.5$. Understanding the effective sample size at $\beta > 1$ would clarify when Route B becomes unreliable and help practitioners choose between routes.

2. How sensitive is Route A to the number of Langevin steps $K$ and the noise schedule $\{\gamma_t\}$? An ablation showing posterior quality as a function of these hyperparameters, especially at extreme $\beta$ values, would help quantify the score-error propagation concern.

**Limitations:**

Yes. Section 6 provides a thorough and honest discussion of route-specific trade-offs, including ESS collapse at extreme $\beta$, support coverage limitations of Route B, score-error amplification in Route A, and sensitivity to architecture and hyperparameters. The impact statement appropriately discusses risks of misuse.

**Strengths And Weaknesses:**

**Strengths:**

- *Well-motivated problem.* Existing GBI methods require running MCMC separately for each $(x_\text{obs}, \beta)$ pair. Amortizing jointly over $x$ and $\beta$ in the SBI setting is practically useful and enables iterating over the tempering parameter.

- *Clean theoretical grounding.* Proposition 4.1 correctly establishes that the SNIS-weighted NPE objective minimizes the forward KL to the power posterior.

- *Two complementary routes with clear trade-off analysis.* Route A (score-based synthesis) can explore off-manifold and recover missing support under misspecification; Route B (SNIS reweighting) avoids MCMC entirely and is cheaper.

- *Strong applied experiment.* The Hodgkin-Huxley experiment (Section 5.2) uses a realistic 8-parameter neuroscience simulator with real data from the Allen Cell Types Database. The posterior marginals are stable across $\beta$ and the predictive traces qualitatively match experimental voltage recordings, demonstrating practical utility under model mismatch.

- *Well-written paper* with clear notation and thorough appendices (benchmark definitions, reference sampler details, architectural choices).

**Weaknesses:**

- *Theoretical coverage gap for $\beta > 1$.* Proposition 4.2 bounds the NRE weight variance only for $\beta \in [1/2, 1]$, but experiments evaluate up to $\beta = 1.5$. For $\beta > 1$, the importance weights $w_\beta \propto \hat{r}^{\beta-1}$ can have unbounded variance — consistent with the observed degradation of Route B in Figure 1 at large $\beta$. This gap between theory and experimental range should perhaps be more explicitly discussed.

- *Noisy evaluation metrics.* The C2ST and MMD curves in Figure 1 are highly volatile across $\beta$, particularly for SLCP and Lorenz-96. Without confidence intervals over test observations, it is difficult to draw firm conclusions about relative method quality. The claim of competitive performance rests largely on visual comparison of noisy curves.

---

> ### Author Rebuttal · Authors · 2026-03-30
>
> 1. Theoretical coverage gap for $\beta > 1$ in Route B
>
> We agree this should be clarified. Proposition 4.2 gives a finite-variance guarantee for the original SNIS-based RouteB only for $\beta \in [1/2,1]$; experiments with $\beta>1$ are stress tests to show where simple SNIS degrades and when RouteA becomes preferable. We will make this theory/empirical distinction explicit and add empirical ESS as a function of $\beta$.
>
> This limitation is not intrinsic to amortized power-posterior inference itself, but rather to the simple proposal used in RouteB. In follow-up analysis, we considered a defensive IS variant with a mixture proposal $p_{\lambda}(\theta,x)=(1-\lambda)\pi(\theta)p(x\mid\theta)+\lambda\pi(\theta)p(x)$, which yields $w_{\beta,\lambda} =
> \frac{r^{\beta}}{(1-\lambda)r+\lambda},
> \qquad
> r=\frac{p(x\mid\theta)}{p(x)}.$
>
> This construction suppresses extreme likelihood-ratio behavior, leading to more stable weights and typically improved ESS in practice. Moreover, our follow-up analysis suggests that under this proposal, the weight variance remains finite throughout the wider regime $\beta\in(0,1]$. We will clarify that the finite-variance guarantee in Proposition 4.2 applies only to the original SNIS construction, while defensive proposals provide a principled route to extending the stable regime of RouteB.
>
> We also add an empirical ESS diagnostic as a function of $\beta$ for RouteB{} (anonymous figure link: \url{https://iili.io/BJrIvII.png}). Here ESS is computed separately for each independent draw using its $K=2000$ importance samples, and we then summarize across 30 draws by reporting the mean and the 25--75th percentile band. The results are consistent with the above interpretation: ESS is highest near $\beta=1$ and degrades as $\beta$ moves away from this regime, with substantially sharper degradation on the harder tasks (SLCP and Lorenz--96) than on the easier ones (Gaussian Mixture and Two Moons). This supports our interpretation that the practical deterioration of simple SNIS reweighting is driven by increasingly unstable importance weights, rather than by a failure of amortization itself.
>
> 2. Noisy C2ST/MMD curves and lack of confidence intervals
>
> The curves in Figure 1 are averaged over multiple held-out datasets, so the reported trends are not driven by any single dataset. We agree, however, that the current presentation would be clearer with explicit uncertainty quantification across datasets. Adding error bars would make the figure easier to interpret. We will revise Figure 1 accordingly and make clear that the reported trends are aggregated over datasets rather than determined by a single example.
>
> Beyond illustrating the route trade-off, Figure 1 is also meant to show that our method achieves posterior quality competitive with existing SBI approaches, while additionally retaining amortization across both datasets and tempering levels. We do not expect the addition of uncertainty quantification to change the qualitative conclusion: RouteB is typically more efficient when reweighting is stable, whereas RouteA becomes more attractive when support mismatch or stronger tempering makes simple reweighting unreliable. Please also see our response to Reviewer RwN4, Point 3, where we discuss this issue in more detail.
>
> 3. Sensitivity of Route A to Langevin hyperparameters
>
> RouteA is more sensitive to the Langevin step sizes than to the number of steps $K$. Although we use separate step sizes for $\theta$ and $x$ together with an annealing schedule, the performance can be sensitive to step size. A likely reason is that RouteA evolves in the joint $(\theta,x)$ space, where the score components for $\theta$ and $x$ can differ substantially in scale. As a result, even with separate step sizes and cooling, a poor initial discretization can still lead to under-shooting in some directions and over-shooting in others.
>
> We will make this mechanism more explicit in the revision, and clarify that RouteA is particularly useful for support recovery or off-manifold exploration, but correspondingly requires more careful tuning of the initial step sizes. We will add the requested ablation.
>
> See also response to reviewer 2NcP section 2.

---

> > ### Author Rebuttal · Reviewer_yt6F · 2026-04-04
> >
> > I thank the authors for their response; the original review I believe addresses the strengths and limitations of the paper, and I would like to keep my positive score.

---

### Decision · Program_Chairs · 2026-04-30

**Decision:**

Accept (regular)

**Comment:**

The average rating is 3.75, with scores (4, 4, 4, 3) leaning toward acceptance. Reviewer RwN4 raised their score from 3 to 4 after the rebuttal, and reviewer yt6F marked concerns as fully resolved. Reviewer tbCp kept a score of 3 but explicitly stated "My concerns have been adequately addressed," with remaining reservations being about experiments being "good but less convincing" rather than any methodological issue.

The core contribution is a single beta-conditioned NPE trained via either score-based synthesis (Route A) or SNIS reweighting (Route B), with a consistency result for the SNIS objective. Reviewer RwN4 characterizes the methods as building on prior work "in a natural way, while making a novel contribution," and 2NcP concludes the work "would be useful for practitioners/researchers interested in amortised beta-annealed inference." Reviewers raised valid concerns about title overclaim (power posteriors vs general GBI), the theory-experiment gap for beta > 1, noisy C2ST curves without confidence intervals, and the qualitative nature of the Hodgkin-Huxley experiment. The authors engaged seriously with all points, moderated their claims ("we do not claim that the amortized methods are uniformly best"), produced new ESS diagnostics, and committed to revising scope wording and Figure 1 presentation. Remaining weaknesses are fixable rather than fundamental. On presentation, the paper would benefit from polishing for the camera-ready version, addressing notation, figure clarity, and prose. For these reasons, the consensus among reviewers is to accept.